# The evolution of antibiotic resistance is associated with collateral drug phenotypes in *Mycobacterium tuberculosis*

Natalie J. E. Waller [1,2], Chen-Yi Cheung[1], Gregory M. Cook [1,2] & Matthew B. McNeil[1,2] ✉

The increasing incidence of drug resistance in *Mycobacterium tuberculosis* has diminished the efficacy of almost all available antibiotics, complicating efforts to combat the spread of this global health burden. Alongside the development of new drugs, optimised drug combinations are needed to improve treatment success and prevent the further spread of antibiotic resistance. Typically, antibiotic resistance leads to reduced sensitivity, yet in some cases the evolution of drug resistance can lead to enhanced sensitivity to unrelated drugs. This phenomenon of collateral sensitivity is largely unexplored in *M. tuberculosis* but has the potential to identify alternative therapeutic strategies to combat drug-resistant strains that are unresponsive to current treatments. Here, by using drug susceptibility profiling, genomics and evolutionary studies we provide evidence for the existence of collateral drug sensitivities in an isogenic collection *M. tuberculosis* drug-resistant strains. Furthermore, in proof-of-concept studies, we demonstrate how collateral drug phenotypes can be exploited to select against and prevent the emergence of drug-resistant strains. This study highlights that the evolution of drug resistance in *M. tuberculosis* leads to collateral drug responses that can be exploited to design improved drug regimens.

Infections from *Mycobacterium tuberculosis* remain a significant global health problem. Fully drug-susceptible strains of *M. tuberculosis* can be treated with a combination of four drugs for six months, that when taken correctly can achieve cure rates of approximately 85%[1]. However, patient non-compliance and suboptimal drug penetration frequently result in treatment failure, driving the evolution and spread of drug resistance in *M. tuberculosis* against all clinically utilized antibiotics[2]. Even for newly approved drugs bedaquiline (BDQ) and pretomanid (PA824) clinical resistance has already been observed, even in treatment-naive populations[3,4]. Drug-resistant strains of *M. tuberculosis* are difficult to treat, requiring longer treatment regimens along with more toxic side effects. This highlights the need for not only new drugs, but rational approaches to the design of combination therapies that can shorten the length of treatment and prevent the emergence of drug resistance.

Studies in various bacterial species and cancer cell lines have demonstrated that the evolution of drug resistance can have collateral drug phenotypes that alter susceptibility (i.e. either sensitization or resistance) to other functionally or structurally unrelated drugs[5–12]. Although there are variations in (i) the reproducibility of evolutionary pathways that produce collateral phenotypes[13], (ii) the effects of different genotypes within the same gene[14] and (iii) the collateral pathways between different species, the exploitation of collateral drug phenotypes holds significant therapeutic promise. For example, drug combinations with overlapping collateral sensitivities could be used to select against the emergence of drug resistance, while drugs that target collateral phenotypes and have increased potency against resistant variants could be used to rapidly eliminate drug-resistant variants and reduce treatment times[6,12,15]. Alternatively, the identification of collateral resistance would identify drug combinations to avoid.

[1]Department of Microbiology and Immunology, University of Otago, Dunedin, New Zealand. [2]Maurice Wilkins Centre for Molecular Biodiscovery, University of Auckland, Auckland, New Zealand. ✉e-mail: matthew.mcneil@otago.ac.nz

There is evidence to suggest that collateral drug phenotypes exist in mycobacterial species. In *M. tuberculosis*, multidrug-resistant strains have increased susceptibility to β-lactams[16], and mutations in the transcriptional regulator *rv0678* result in cross-resistance between BDQ and clofazimine (CFZ)[17]. In *Mycobacterium smegmatis* low-level resistance to fluoroquinolones produces resistance to a range of tuberculosis (TB) therapeutics, including isoniazid (INH) and ethionamide (ETH)[18]. Although many small molecules have been tested for activity against drug-resistant clinical isolates of *M. tuberculosis*, inferences of collateral drug phenotypes are complicated by the lack of an isogenic drug-sensitive parent strain to which changes in susceptibility can be appropriately compared.

Here, we sought to address this gap in knowledge by identifying the collateral effects of drug resistance in *M. tuberculosis*. To overcome the lack of an isogenic reference strain, we evolved resistant strains of *M. tuberculosis* against a panel of 23 clinically relevant drugs with diverse structures and mechanisms of action. Susceptibility profiles of resistant mutants were determined and relationships between genotype and phenotype were established. Examples of collateral drug sensitization were identified, with most collateral drug sensitizations being uni-directional and in some cases conserved among genetically diverse resistant backgrounds. We also identified examples of cross-resistance, and further expanded the pharmacophores subject to efflux by mutations in *rv0678*. Finally, combinations of antibiotics with overlapping collateral phenotypes were able to selectively eliminate resistant strains from a mixed population and were able to delay and in some cases prevent the emergence of drug resistance. These results highlight the therapeutic potential of exploiting collateral drug phenotypes to drive the design of novel treatment regimens against *M. tuberculosis*.

## Results

### Collateral drug phenotypes of drug-resistant *M. tuberculosis*

To identify collateral drug phenotypes in *M. tuberculosis*, we experimentally evolved resistant strains of *M. tuberculosis* strain mc²6206 (Δ*leuCD*, Δ*panCD*) to increasing concentrations (i.e. 3, 10, and 30× the minimum inhibitory concentration [MIC]) of a panel of 23 clinically relevant drugs that target a diverse array of cellular pathways (Table 1). *M. tuberculosis* strain mc²6206 is an avirulent auxotrophic strain derived from H37Rv that is approved for use under PC2 conditions at the University of Otago, mitigating the risk of working with virulent strains that require PC3 containment. *M. tuberculosis* strain mc²6206 behaves identically to H37Rv when grown in supplemented media[19–21]. We were able to isolate resistant mutants on solid media against all selected compounds for at least a single concentration used. Except for mutants isolated against nitrofurantoin (NFT) and SQ109, all mutants showed an increase in MIC against their parent compounds when validated in liquid MIC assays (Fig. 1 and Supplementary Dataset 1). Mutants raised against ethambutol (EMB), had significant growth defects in 96-well plates. For these reasons, SQ109, NFT, and EMB-raised mutants were excluded from further analysis. Where possible, three resistant mutants raised against a single compound were selected for further analysis, including antimicrobial susceptibility profiling against all 23 compounds and whole genome sequencing (WGS) to identify the most probable mutation responsible for resistance to the isolating compound (Fig. 1). For strains with resistance to clinically utilized anti-tubercular agents including rifampicin (RIF), fluoroquinolones (i.e. levofloxacin (LEV)), aminoglycosides (i.e. capreomycin (CAP), kanamycin (KAN) and streptomycin (STRP)) and linezolid (LZD), our isolated mutations are consistent with those observed in clinical isolates of in *M. tuberculosis*[22]. Strains with resistance to INH did not contain the dominant genotype (i.e. KatG^S315T) that is seen in the majority of clinical isolates[23]. This is consistent with prior reports of in vitro isolated INH resistant mutants[24]. For some resistance loci that require inactivation to generate resistance (i.e. *ddn*, *fbiABC*,

## Table 1 | Compounds used in this study

| Antibiotic | Abbreviation | Target | Liquid MIC against *M. tuberculosis* mc²6206 susceptible parent strain (μM) |
|---|---|---|---|
| Rifampicin | RIF | Transcription | 0.1 |
| Fidaxomicin | FIX | Transcription | 4 |
| Linezolid | LZD | Translation | 3 |
| Kanamycin | KAN | Translation | 4 |
| Streptomycin | STRP | Translation | 0.15 |
| Capreomycin | CAP | Translation | 1.1 |
| Levofloxacin | LEV | DNA gyrase | 0.8 |
| Bedaquiline | BDQ | Bioenergetics | 1 |
| Clofazimine | CFZ | Bioenergetics | 0.5 |
| Q203 | Q203 | Bioenergetics | 0.01 |
| TB-47 | TB47 | Bioenergetics | 0.003 |
| Pretomanid | PA824 | Bioenergetics/ cell wall | 0.8 |
| Isoniazid | INH | Cell wall | 0.5 |
| Ethionamide | ETH | Cell wall | 10 |
| Ethambutol | EMB | Cell wall | 3 |
| PBTZ-169 | PBTZ | Cell wall | 0.001 |
| AZ7371 | AZ7 | Cell wall | 1 |
| SQ109 | SQ109 | Cell wall | 0.8 |
| Thioacetazone | TAC | Cell wall | 4 |
| Thiocarlide | TCL | Cell wall | 5 |
| Thiophene-2 | THP | Cell wall | 1.5 |
| Compound-1 (Molecule #4[29]) | CPD1 | Cell wall | 0.1 |
| Tunicamycin | TUN | Cell wall | 1.5 |
| Nitrofurantoin | NFT | Undefined | 50 |

*rv0678*), available clinical data shows that there is a large spectrum of mutations[22]. While most mutations identified in these genes in this study have not yet been observed clinically, the loss-of-function genotype does not preclude them from eventually being observed. Furthermore, some compounds have yet to be utilized in a clinical setting so clinically relevant resistance loci have yet to be defined. Secondary mutations identified in all strains are described in Supplementary Dataset 2. As compounds were prepared along a three-fold dilution series in minimum inhibition concentration (MIC) assays, we defined collateral changes in drug susceptibility as a three-fold change (i.e. resistance or sensitization) in MIC relative to the drug-susceptible mc²6206 parental strain (Fig. 1 and Supplementary Dataset 1). As we have utilized a three-fold dilution series, strains that had a reproducible 2-3-fold change in MIC were considered to have a potentially low-level change in susceptibility. We observed several cases of uni-directional collateral sensitization and bi-directional collateral resistance between functionally distinct drugs. We also observed several strains with smaller 2–3-fold shifts in MIC (Fig. 1), some of which were consistent with the prior results, e.g. low-level resistance of ETH resistant *mshA* and *mshC* mutants against INH (Fig. 1)[25].

In conclusion, our results suggest that drug resistance in *M. tuberculosis* is associated with collateral drug phenotypes.

### Collateral drug-resistant phenotypes associated with drug resistance in *M. tuberculosis*

Cross resistance was most frequent among drugs with similar mechanisms of action. For example, strains selected for resistance against the cytochrome *bc₁:aa₃* inhibitor Q203, were cross resistant to an alternative cytochrome *bc₁:aa₃* inhibitor TB47 due to overlapping

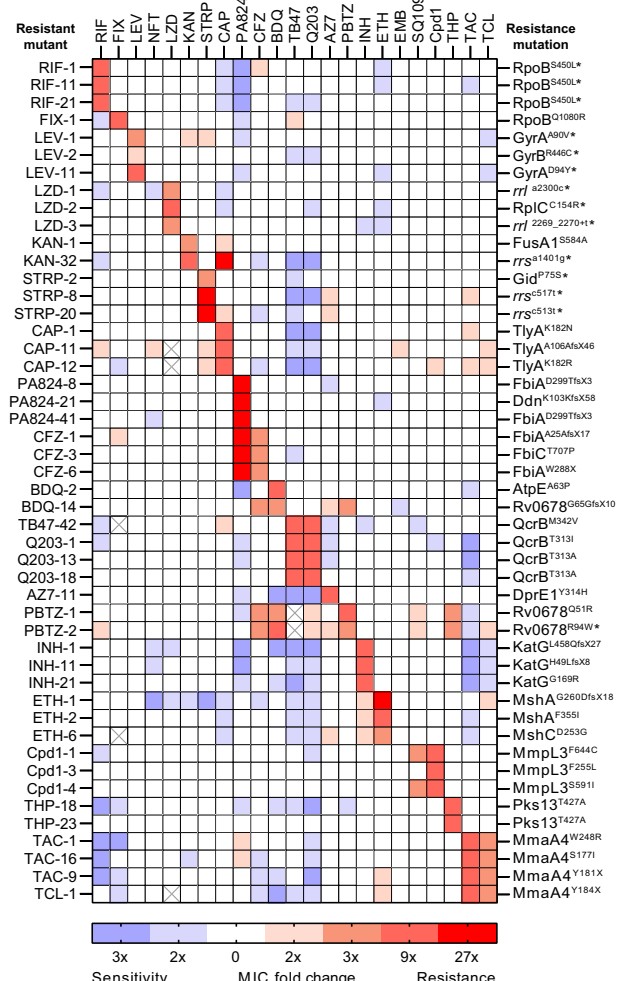

**Fig. 1 | Collateral phenotypes associated with drug resistance in *M. tuberculosis*.** Heatmap represents the relationships of collateral phenotypes (i.e. both resistance and sensitization) as determined by changes in minimum inhibitory concentrations (MIC) for 48 antibiotic-resistant variants evolved against 23 antibiotics. Colour coding represents the average fold increase (red) or decrease (blue) in MIC value for each resistant variant relative to the drug-susceptible parent. For each strain and antibiotic combination, a minimum of four independent experiments comparing the MIC of each drug-resistant strain relative to the drug-susceptible parent were used. Strain names are on the left and antibiotics used in drug-susceptibility profiling are along the top. Resistance mutations hypothesized as being responsible for the primary drug resistance are listed on the right and classified as previously defined[85]. Briefly, an 'X' in place of an amino acid after the codon position indicates a stop codon. A 'fsX' indicates a frameshift resulting in a stop codon at the defined number of amino acids into the new reading frame. '+' indicates a nucleotide insertion at specified location. Asterisks (*) represent mutations which have been observed in clinical isolates. Crossed out cells represent no data, due to antibiotic availability issues. RIF = rifampicin, FIX = fidaxomicin, LEV = levofloxacin, LZD = linezolid, KAN = kanamycin, STRP = streptomycin, CAP = capreomycin, PA824 = pretomanid, CFZ = clofazimine, BDQ = bedaquiline, AZ7 = AZ7371, PBTZ = PBTZ-169, INH = isoniazid, ETH = ethionamide, EMB = ethambutol, THP = thiophene-2, TAC = thioacetazone, TCL = thiocarlide.

mutations in *qcrB*[26,27] (Fig. 1 and Supplementary Fig. 1a). There was also cross resistance between (i) MmpL3 inhibitors (i.e. SQ109[28] and CPD1[29]) with resistant strains containing mutations in *mmpL3* and (ii) inhibitors of the HadABC dehydratase complex (i.e. thioacetazone [TAC] and thiocarlide [TCL]) with resistant strains having mutations in *mmaA4*[30–34] (Fig. 1 and Supplementary Fig. 1b–e). Consistent with prior reports, cross resistance between BDQ and CFZ was mediated by mutations in the negative transcriptional regulator *rv0678* that

controls the expression of the *mmpL5* efflux pump[17] (Figs. 1 and 2a, b). Mutations in *rv0678* also resulted in resistance to PBTZ-169, a DprE1 inhibitor currently in phase 2 trials (Fig. 2c). Consistent with this, *rv0678* mutants were selected for in the presence of PBTZ-169 (Fig. 1). These observations validate recent chemical-genetic screens investigating resistance mechanisms to PBTZ-169[35]. Mutations in *rv0678* also provided low-level resistance (i.e. 2–3-fold) to Q203 and AZ7371 (AZ7) (Fig. 1).

Interestingly, strains selected for low-level resistance (i.e. 2–3-fold) to CFZ had cross resistance to PA824, yet PA824 resistant mutants had a less than 3-fold shift in MIC to CFZ (Figs. 1 and 2d and Supplementary Fig. 1f). CFZ-resistant strains frequently have mutations in *rv0678*[17], while resistance to PA824 is associated with the inactivation of genes involved in F420 biosynthesis (*fbiA*, *fbiB*, *fbiC*, *fbiD*, *ddn*) and the F420-dependent glucose-6-phosphate dehydrogenase (*fgd1*) of the pentose-phosphate pathway[36,37]. Interestingly, WGS demonstrated that CFZ-resistant strains had mutations in *fbiC* and *fbiA* (Fig. 1). On this basis, we hypothesized that the inactivation of specific PA824 resistance genes provides low-level resistance to CFZ. To investigate this, we transcriptionally repressed PA824 resistance genes using mycobacterial CRISPR interference (CRISPRi)[38,39] and determined their sensitivity to CFZ using a two-fold dilution series to account for potentially smaller shifts in MIC. The non-targeting sgRNA control had MIC's for PA824 and CFZ of ~0.4 μM and 0.3 μM respectively, comparable to the parental strain (Fig. 2a, d, e, f). Consistent with a role in PA824 resistance, all transcriptionally depleted strains had resistance to PA824 compared the non-targeting control (Fig. 2e). Interestingly, repression of *ddn* provided the largest increase in MIC, with a 46-fold increase, followed by *fbiD* and *fgd1* (~40× increase), *fbiA* and *fbiC* (~16–29× increase) and *fbiB* (~5× increase). Consistent with resistant mutants, the repression of *ddn* provided no change in susceptibility to CFZ (i.e. an MIC of 0.25 μM), while the repression of *fbiA*, *fbiB*, and *fbiC* provided low-level resistance to CFZ (i.e. ~1.8–2.4-fold increase) (Fig. 2f). The repression of *fbiD* and *fgd1* provided a less than two-fold increase in MIC to CFZ (i.e. ~1.7–1.9×) (Fig. 2f). Low-level resistance phenotypes correlated to changes in bacterial killing, with the repression of *fbiA*, *fbiB*, *fbiC*, *fbiD* or *fgd1* resulting in increased cellular viability at concentrations equal to the MIC of the non-targeting control (~1–2 log increase in CFU at 1x MIC) (Fig. 2g, h). All knockdown strains were resistant to killing by PA824 (Supplementary Fig. 2).

In conclusion, while collateral drug resistance in *M. tuberculosis* is most frequently associated with cross resistance between drug classes that have a shared mechanism of action, we have demonstrated that efflux pumps and changes in mycobacterial metabolism provide low-level resistance to diverse pharmacophores.

## Collateral drug sensitivity associated with drug resistance in *M. tuberculosis*

Examples of collateral drug sensitization in *M. tuberculosis* identified in this study were uni-directional (i.e. resistance to drug A produced sensitivity to drug B but not vice versa) (Fig. 1). For example, INH resistant strains with mutations in *katG* had increased sensitivity to cytochrome *bc1* inhibitors (i.e. Q203 and TB47[40]), the ATP synthase inhibitor BDQ[41] and the HadABC dehydratase inhibitor TAC[34] (Figs. 1 and 3a–c). Strains with resistance to Q203, TB47, BDQ or TAC had no cross sensitization to INH (Fig. 1). Furthermore, INH, BDQ, and RIF resistant strains with mutations in *katG*, *atpE*, or *rpoB* respectively, all had increased sensitization to PA824, while PA824 resistant strains had no increased sensitivity to either INH, BDQ or RIF (Figs. 1 and 3d). A fidaxomicin (FIX) resistant mutant with an alternative *rpoB* mutation also had a low-level increase in sensitivity to PA824 (Fig. 1 and Supplementary Fig. 3e). In addition to the INH resistant *katG* mutant, strains with diverse genetic backgrounds, such as mutations in *rrs* (i.e. KAN and STRP resistant) and *dprE1* (i.e. AZ7-resistant), also exhibited

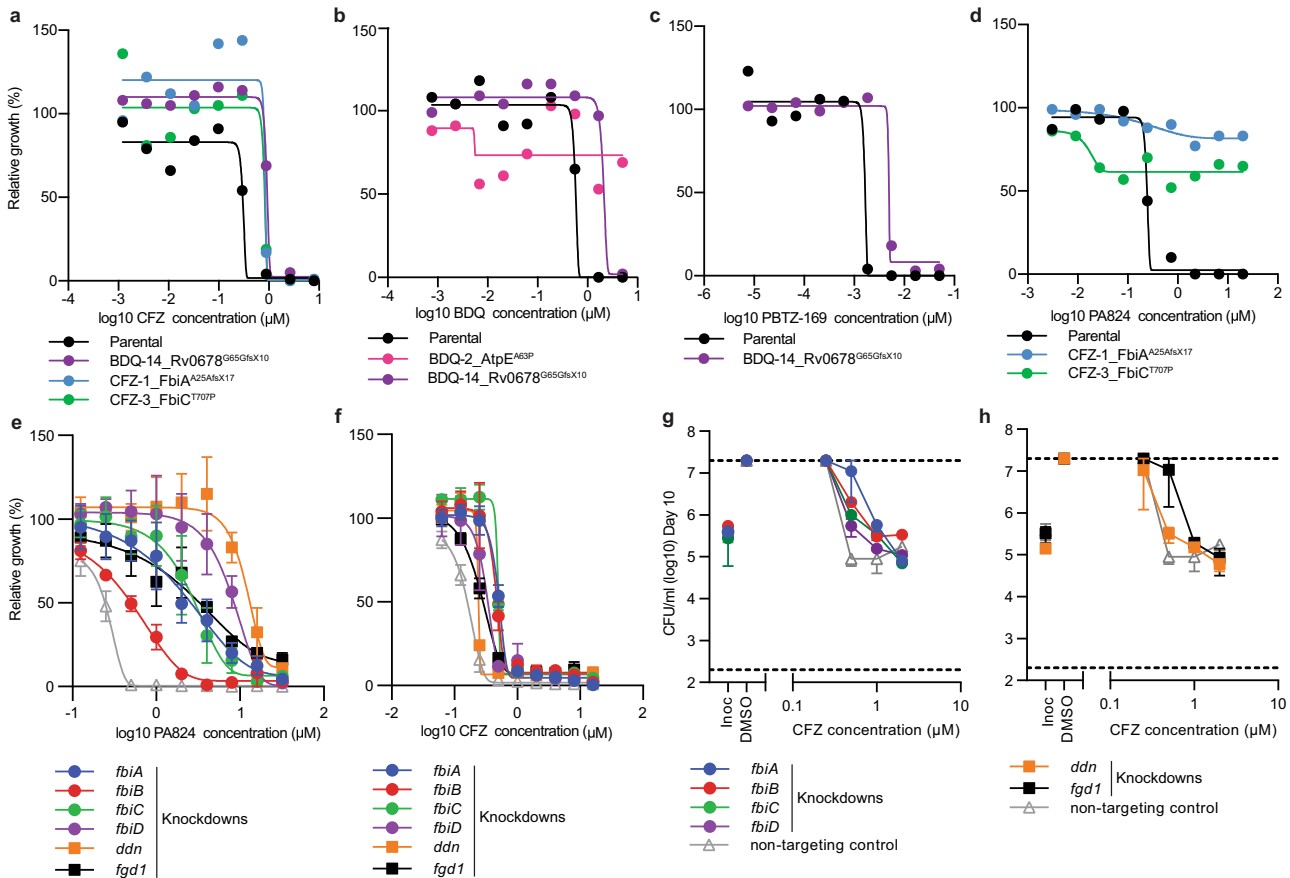

**Fig. 2 | Collateral drug resistance in drug-resistant variants of *M. tuberculosis* is mediated via diverse mechanisms. a–d** Dose–response curves for selected drug-resistant variants and the drug-susceptible parent against selected antibiotics (highlighting collateral drug resistance). Strain names and candidate mutations are listed below each dose-response. **e, f** Relative growth of CRISPRi knockdown and non-targeting control strains against PA824 and CFZ. **g, h** Minimum bactericidal concentration assays for CRISPRi knockdown and non-targeting strains against CFZ. CFUs were determined at day 0 and at day 10. Inoc = starting inoculum on day 0, DMSO = solvent control. Dose–response curves (**a–d**) are the results of a single biological replicate from a representative experiment ($n > 4$ independent experiments). **e–h** are the mean and range of biological duplicates from a representative experiment ($n = 2$ independent experiments). **a–f** The Gompertz model was used to fit growth data. **g, h** Dashed lines represent the upper and lower limits of detection.

an increased sensitivity to Q203 and TB47 (Figs. 1 and 3a). In some instances, differences in the predicted effects of mutations on gene function (i.e. partial, or complete loss) resulted in differing collateral drug phenotypes. For example, a frameshift mutation in MshA (i.e. ETH-1 *mshA*^G260DfsX18^) but not a point mutation in MshA (i.e. ETH-2 *mshA*^F355I^) resulted in increased sensitivity to NFT (Fig. 1 and Supplementary Fig. 3a). Mutations in shared pathways also had different collateral drug phenotypes, with a point mutation in MshC (i.e. ETH-6 *mshC*^D253G^), that with MshA plays a role in mycothiol synthesis, having no increased sensitivity to NFT (Fig. 1 and Supplementary Fig. 3b).

Drug-resistant strains isolated in this study had a unique collection of secondary mutations (Supplementary Dataset 2). Many secondary mutations were conserved among resistant mutants each with separate resistance/sensitivity profiles that were isolated from the same culture that was used in resistant mutant isolation (Supplementary Dataset 2). Consequently, we hypothesized that the majority of secondary mutations would not influence the collateral phenotypes we observed. For example, KAN-1, PBTZ-1 and INH-21 all harbour the same point mutation in PpsB (I896N), a gene theorized to be involved in the biosynthesis of phthiocerol dimycocerosate (PDIM)[42]. However, KAN-1 does not have increased sensitivity to any other antibiotic, and PBTZ-1 only shares two of the five increased sensitivities experienced by INH-21. INH-1 and INH-11 that have comparable collateral profiles to INH-21 had no mutations in PpsB. In some instances, secondary mutations did influence collateral phenotypes as resistant mutants with identical

primary drug-resistance mutations had different responses to drug challenge (Fig. 1 and Supplementary Dataset 2). For example, two strains with resistance to the Pks13 inhibitor thiophene-2 (THP)[43] (i.e. THP-18 and THP-23) both harboured the same point mutation (*pks13*^T427A^), yet had different sensitivities to several antibiotics. THP-18 harboured a mutation in YrbE2A, a predicted ABC transporter permease, that was not observed in THP-23, or any other mutant isolated used in this study. Consistent with the THP-18 collateral sensitivities, chemical genetic studies have demonstrated the loss of YrbE2a increased sensitivity to RIF and BDQ[44] (Fig. 1 and Supplementary Fig. 3c, d). THP-18 also had a low-level sensitivity (i.e. 2-3-fold) to other antibiotics including Q203, PA824 and PBTZ-169 (Fig. 1).

In conclusion, specific mechanisms of drug resistance in *M. tuberculosis* are associated with collateral drug sensitization.

## Collateral drug sensitivities can be exploited to impair the growth of drug-resistant *M. tuberculosis*

To validate results from micro-titre plate assays, we monitored the growth of selected drug-resistant strains in the presence of sub-inhibitory concentrations of drugs to which they had increased sensitivity. Strains were initially challenged with sub-inhibitory, rather than inhibitory concentrations, to highlight differences in drug sensitivity of resistant strains relative to the drug susceptible parent strain. When grown in the presence of 0.3× MIC of the cytochrome $bc_1$:$aa_3$ inhibitor Q203, all tested mutants had an impaired growth compared

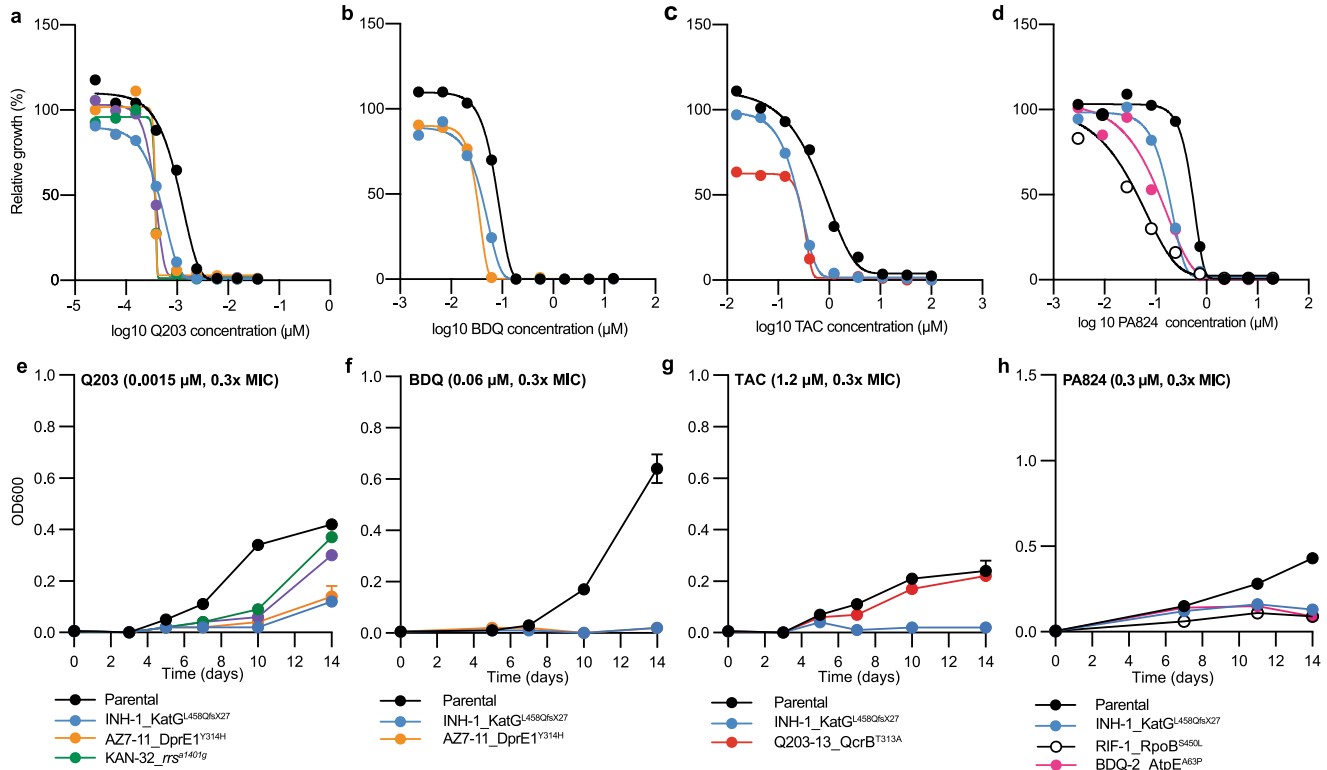

**Fig. 3 | Collateral drug sensitivities can be targeted to limit the growth of drug-resistant variants of *M. tuberculosis*. a–d** Dose–response curves for selected drug-resistant variants and the drug-susceptible parent against selected antibiotics. Strain names and candidate mutations are listed below each dose-response. Dose-response curves are the results of a single biological replicate from a representative experiment (*n* > 4 independent experiments). The Gompertz model was used to fit growth data. **e–h** Time kill assays of selected drug-resistant variants and drug-susceptible variants against sub-inhibitory concentrations (i.e. 0.3× MIC of drug-susceptible parent strain). OD600 data is the mean ± range of biological duplicates from a representative experiment (*n* > 2 independent experiments). Assays were started at an OD600 of 0.005, and OD600 was determined on stated days. DMSO solvent control graphs are included in Supplementary Fig. 5.

to the isogenic parent. In the presence of 0.3× MIC Q203, the growth of the isogenic parent was detected by optical density at day 5, while growth of all tested drug-resistant strains was not detected until or after day 10 (Fig. 3e). The growth of the INH and AZ7-resistant mutants in the presence of 0.3× MIC of BDQ was prevented, with no detectable growth after 14 days, while growth of the isogenic parent was detected by optical density at day 10 (Fig. 3f). Similarly, growth of the INH-resistant mutant was prevented in the presence of 0.3× MIC of TAC, with no detectable growth after 14 days, although the Q203-resistant mutant did not reproduce the collateral phenotype that was observed in microtitre plates (Fig. 3g). The growth of the INH, RIF and BDQ-resistant mutants was impaired in the presence of 0.3× MIC of PA824, with the parental strain detected after day 7, while resistant mutants did not grow (Fig. 3h). In conclusion, sub-inhibitory concentrations of compounds can be used to target collateral drug phenotypes and prevent or impair the growth of drug-resistant strains.

## Collateral drug phenotypes alter the bactericidal properties of antibiotics against drug-resistant strains of *M. tuberculosis*

We hypothesized that changes in susceptibility would correlate with changes in bacterial killing by (1) altering the concentration of compound required to induce bacteriostasis or cidality and/or (2) alter the killing kinetics (i.e. time required and level of reduction in CFU/mL) of drug-resistant strains at concentrations of drugs above the MIC of the isogenic parent strain. To investigate this, we performed MBC assays for selected drug-resistant strains in the presence of compounds to which they had increased sensitivity. Cell viability was determined after 10-day incubation with the drug of interest. AZ7, KAN and STRP mutants, all had an increased sensitivity to sub-inhibitory concentrations to Q203, but MBC results for concentrations above the MIC

displayed no change in CFU/mL, comparable to the isogenic parent (Fig. 4a). This is consistent with Q203 being a bacteriostatic drug. Interestingly, the INH resistant *katG* mutant was killed by concentrations of Q203 above the MIC, reaching the lower limit of detection (i.e. >2.5 log reduction in CFU/mL) at concentrations >3× MIC of Q203 (Fig. 4a). The INH-resistant *katG* mutant was also more susceptible to killing by BDQ, reaching the lower limit of detection (i.e. >2.5 log reduction in CFU/mL) at concentrations >3× MIC of BDQ. The isogenic parent and the AZ7-resistant mutant had a concentration dependent BDQ killing phenotype and reached a maximum of 2-log reduction at 9 and 27× MIC (Fig. 4b). Similarly, TAC which was bacteriostatic against the isogenic parent, was able to kill the INH-resistant *katG* mutant with a >2 log reduction in CFU/mL at concentrations ≥3× MIC of TAC (Fig. 4c). PA824 had a bacteriostatic phenotype against the isogenic parent yet had increased lethality against INH, RIF and BDQ-resistant mutants and achieved a >2 log reduction in CFU/mL at 9 and 27x the MIC (Fig. 4d).

Consistent with MBC results, exposure to 9× the parental MIC of Q203 or TAC in time kill assays, both of which were bacteriostatic against the isogenic parent, killed the INH-resistant mutant reaching the lower limit of detection (i.e. >2.5 log reduction in CFU/mL) within 7 and 10 days, respectively (Fig. 4e, g). Similarly, at 9× MIC BDQ had a time-dependent killing of the isogenic parent, reaching a 2-log reduction after 14 days, while the INH-resistant *katG* mutant reached the lower limit of detection (i.e. >2.5 log reduction in CFU/mL) within 5 days (Fig. 4f). At 9× MIC of PA824, the growth of the isogenic drug-susceptible parent was suppressed for approximately 10 days, after which PA824-resistant clones began to grow achieving a >1 log increase in CFU/mL by day 14 (Fig. 4h). Consistent with MBC results the INH, RIF and BDQ-resistant mutants were killed by 9× MIC of PA824, achieving

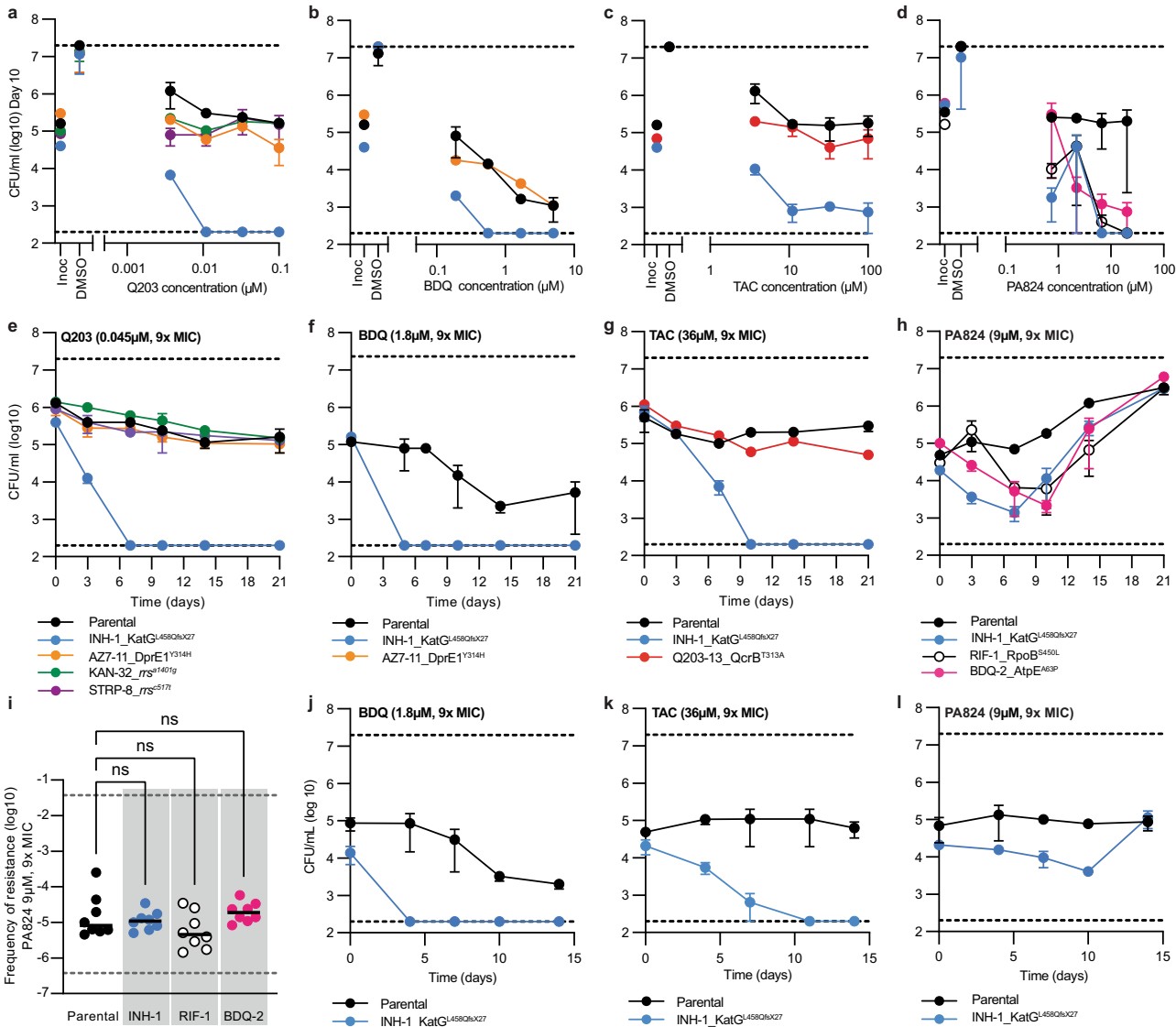

**Fig. 4 | Altered killing dynamics of drug-resistant variants of *M. tuberculosis* following exposure to collateral antibiotics. a–d** Minimum bactericidal concentration assays for selected drug-resistant variants and the drug-susceptible parent against selected antibiotics. CFUs were determined at day 0 and at day 10 for concentrations at and above the minimum inhibitory concentrations (i.e. 1, 3, 9, and 27× MIC of the drug-susceptible parent) of selected compounds. Inoc = starting inoculum on day 0, DMSO = solvent control. **e–h** Time kill experiments of selected drug-resistant variants and the drug-susceptible parent against selected antibiotics at concentrations above the minimum inhibitory concentration (9x MIC). CFUs were taken at stated time points. **i** Frequency of resistance for selected drug-resistant mutants and drug-susceptible parental strain against 9× PA824. Statistical significance was investigated using a one-way ANOVA and corrected for multiple comparisons using the Dunnett's multiple comparisons test (95% CI). ns = not significant. **j–l** Co-culture time kills. Drug-susceptible parent strain and INH-1 were combined at 1:1 ratio and challenged with BDQ, TAC or PA824 above minimum inhibitory concentrations (9× MIC). CFUs were taken at stated time points and plated on 7H11 with/without 9× MIC INH to determine the proportion of INH-1 in the co-culture relative to the drug-susceptible parent. For **a–h**, data are the mean ± range of biological duplicates from a representative experiment (*n* > 2 independent experiments). For **i**, data are the mean of eight biological replicates. For **j–l**, data are the mean ± range of biological duplicates from a representative experiment (*n* > 2 independent experiments). Strain names and resistance mutations are listed. Dashed lines represent upper and lower limits of detection. DMSO solvent control graphs are included in Supplementary Fig. 5.

between a 1–2 log reduction in CFU/mL by day 7. Interestingly, by day 14 all mono-resistant strains developed secondary drug resistance to PA824, as demonstrated by an increase in CFU/mL from their greatest reduction in CFU/mL (Fig. 4h). By day 21, viable colonies were comparable to the isogenic drug-susceptible parent (Fig. 4h).

We hypothesized that the difference in kill kinetics and the delayed emergence of resistance against PA824 for drug-resistant strains was the result of a reduced ability to evolve secondary resistance to PA824 compared to the isogenic drug-susceptible parent. To investigate this, we determined the frequency of resistance of the isogenic parent and the BDQ, RIF and INH mutants against 9× the MIC

of PA824. Consistent with published results the in vitro frequency of resistance for the isogenic parental strain against 9× MIC PA824 was $4.37 \times 10^{-5}$ (Fig. 4I)[36]. The frequency of resistance for each resistant mutant was comparable with the parental strain (Fig. 4I). This demonstrates that the BDQ, RIF and INH mutants have increased sensitivity to killing by PA824, but that this does not alter their ability to evolve secondary drug resistance to PA824.

In conclusion, collateral drug phenotypes in drug-resistant strains of *M. tuberculosis* can alter the killing dynamics of antibiotics, yet in some cases does not limit their ability to evolve secondary drug resistance.

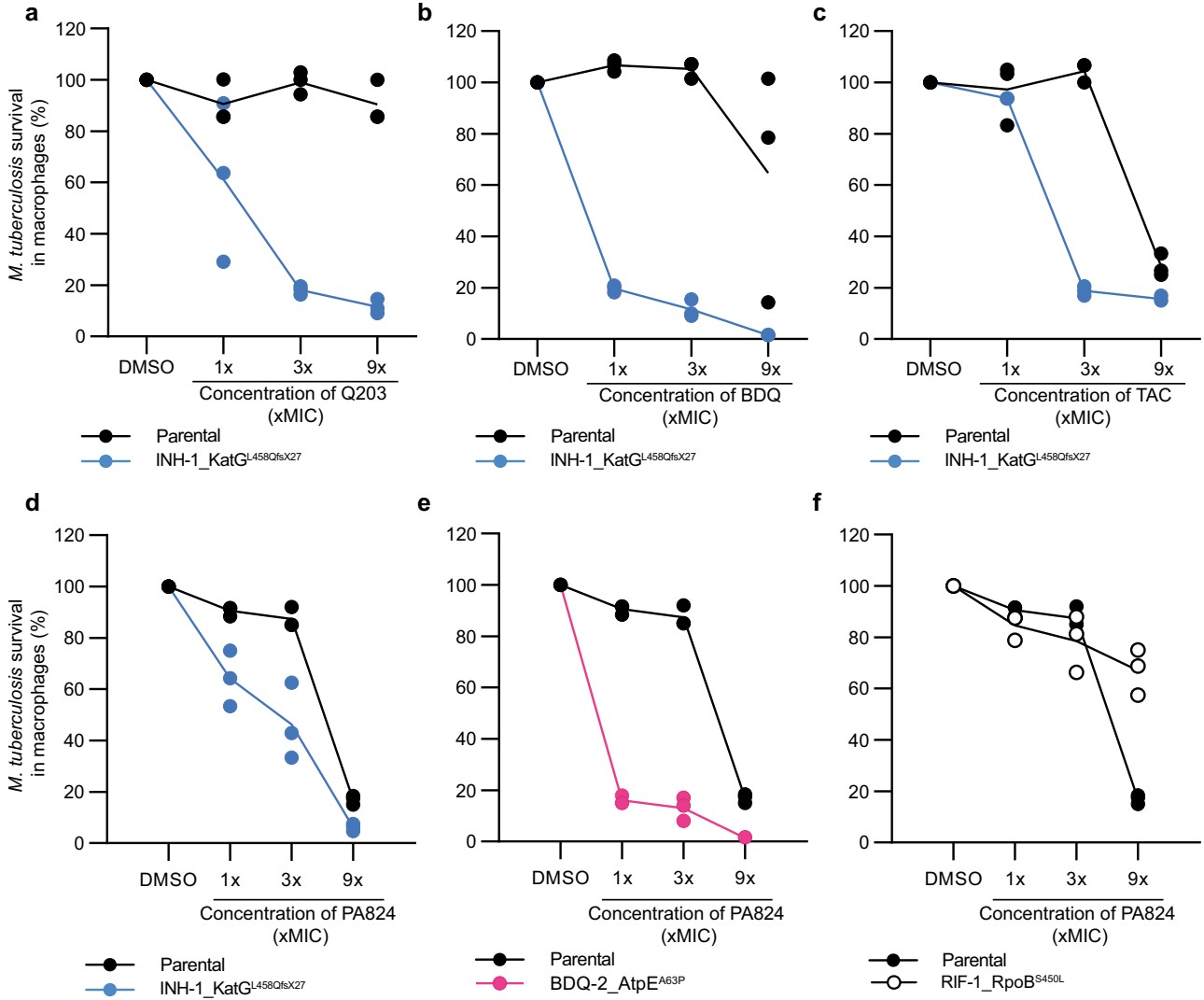

**Fig. 5 | Collateral drug sensitivities are retained within infected THP-1 macro-phages. a–f** The survival of intracellular *M. tuberculosis* drug-susceptible or resistant strains within THP-1 macrophages was determined after 3 days of exposure to compounds at or above the MIC. MICs are as follows; Q203 = 0.05 μM, BDQ = 0.2 μM, TAC = 0.5 μM, PA824 = 1 μM. CFUs/mL were used to determine *M. tuberculosis* survival. Results are presented as percentage (%) survival relative to the no com-pound DMSO control. Line graphs represent the mean of three biological repli-cates, with individual replicate values plotted for each concentration. Results are from a representative experiment (*n* = 2 independent experiments).

## Collateral phenotypes can be targeted to select against drug-resistant strains in a mixed population

We hypothesized that the collateral drug phenotypes could be exploited to selectively eliminate drug-resistant strains of *M. tuberculosis* from a heterogenous population. To investigate this, we mixed the drug-susceptible parent strain and the INH resistant *katG* mutant (i.e. INH-1) at a 1:1 ratio and challenged the population with 9× the parental MIC of either BDQ, TAC or PA824 and monitored viability over time. When exposed to BDQ, the parental strain presented a ~1.5 log reduction in CFU by day 14, whereas INH-1 was cleared from the population (~2.5 log reduction in CFU) by day 4 (Fig. 4j). Similar results were seen with TAC which reduced the number of INH-1 CFU by ~2 logs by day 11, while the parental strain exhibited a bacteriostatic response (Fig. 4k). Treatment with PA824 displayed a bacteriostatic response from the parental strain with minimal change in CFU from day 0 to day 14. The INH-1 mutant had an initial decrease in CFU by day 10 (<1 log reduction), followed by a subsequent increase due to the outgrowth of mutants with secondary resistance (Fig. 4l).

In conclusion, collateral sensitivities can be targeted in a hetero-genous population to select against drug-resistant strains.

## Collateral sensitivities can be exploited under host relevant conditions

To investigate whether collateral drug phenotypes could be exploited under host relevant conditions, we performed THP-1 macrophage infection studies with the parental strain and INH-1. Q203 was bac-teriostatic against drug-susceptible *M. tuberculosis*, yet reduced INH-1 viability by >80% at 3 and 9× MIC (Fig. 5a). Similar results were observed with BDQ, which reduced viability by 80 and 89% at 1 and 3× the MIC, concentrations that had no effect on the survival of drug-susceptible *M. tuberculosis* (Fig. 5b). At 9× MIC of BDQ, the viability of INH-1 was reduced by 98% while the drug-susceptible was reduced by an average of 36% (Fig. 5b). TAC also had increased potency against the INH-1 mutant, although the increased killing was only observed at 3× MIC, reducing the viability of INH-1 by 82%, yet had no effect against drug-susceptible *M. tuberculosis* (Fig. 5c).

Multiple genetically distinct strains, i.e. INH-1, RIF-1 and BDQ-2, had a shared collateral sensitivity to PA824 in in vitro conditions. These collateral phenotypes could also be exploited in host-relevant conditions, with both INH-1 and BDQ-2 showing increased killing relative to the parent strain (Fig. 5d, e). Specifically, at 1× and 3× the MIC the parental strain exhibited a ~10% decrease in viability while INH-1 was reduced by ~40 and 60%, and BDQ-2 was reduced by ~80% (Fig. 5d–f). At 9× MIC, viability of the parental strain was reduced by ~80%, while INH-1 and BDQ-2 were both reduced by >90% (Fig. 5d, e). Interestingly, RIF-1 had comparable viability to the parental strain at 1 and 3× MIC, yet had increased tolerance at 9× the MIC, with a ~30% in viability compared to the parental strain that was reduced ~80% (Fig. 5f).

In conclusion, collateral drug-sensitivities of drug-resistant strains identified under in vitro conditions can be exploited under host relevant conditions, although the extent of viability reduction varies between resistant strains and drug concentrations.

## Combination therapies that target collateral phenotypes can suppress the evolution of drug resistance in *M. tuberculosis*

We hypothesized that drug combinations that exploit collateral drug phenotypes would delay or prevent the emergence of drug-resistance. To investigate this, we monitored the emergence of INH-resistant mutants by combining INH at 9× MIC with sub-inhibitory concentrations (i.e. 0.3× MIC) of compounds that impaired or prevented the growth of an INH-resistant *katG* mutant, specifically BDQ, Q203, PA824 and TAC (Fig. 3e–h). In time kill assays, and consistent with prior reports, when exposed to 9× MIC of INH drug-susceptible *M. tuberculosis* was initially killed with a ~1.5-log reduction in CFU/mL, followed by the emergence of resistant clones around day 10 producing a 'V' shaped viability curve (Fig. 6a–d)[45]. Consistent with prior reports, sub-inhibitory concentrations of Q203 antagonized INH-killing, with a maximum ~1.5-log reduction in CFU/mL not being reached until 14 days, compared to only 7 days needed with INH alone (Fig. 6a)[46]. Despite this antagonism, sub-inhibitory concentrations of Q203 delayed the out-growth of INH-resistant mutants, with an increase in CFU/mL not being detected until day 21 (Fig. 6a). Again, consistent with prior reports, sub-inhibitory concentrations of BDQ also antagonized INH-killing with no viable reduction in CFU/mL for 14 days (Fig. 6b)[46]. Despite this antagonism, addition of sub-inhibitory concentrations of BDQ with 9× INH delayed the out-growth of INH-resistant mutants, with an increase in viable cells only being detected at day-21 (Fig. 6b). Sub-inhibitory concentrations of TAC did not alter the rate of INH killing with colonies reaching similar reductions in CFU/mL as INH alone at 7 days (Fig. 6c). Interestingly, sub-inhibitory concentrations of TAC promoted continued INH mediated killing, reaching the lower limit of detection by day 10 (i.e. >2-log reduction in CFU/mL) (Fig. 6c). Viable colonies remained undetected until day 35, when outgrowth of a single replicate culture was detected (Fig. 6c). Viable colonies remained undetected in the second replicate (Fig. 6c). This development of resistance in only one of the technical replicates was seen across multiple experiments, although there was variation in the time taken for this resistance to develop (Supplementary Fig. 4). Sub-inhibitory concentrations of PA824 delayed INH killing, reaching ~1.5-log reduction in CFU/mL after 10 days compared to 7 days for INH alone (Fig. 6d). Sub-inhibitory concentrations of PA824 also delayed the outgrowth of INH-resistant mutants, with an increase in CFU/mL not detected until day 28 (Fig. 6d).

Although all compounds tested (i.e. BDQ, Q203, PA824 and TAC) impaired the growth of an INH-resistant *katG* mutant (Fig. 3a–d) we hypothesized that the variations in the delayed out-growth of INH-resistant mutants in drug-combination studies was a result of differences in the effects of sub-inhibitory concentrations on the viability of INH-resistant *katG* mutants. Sub-inhibitory concentrations of Q203 and PA824, each of which had increasing delays on the emergence of INH-mutants, extended the lag phase of the INH-resistant mutant when measured by CFU/mL (Fig. 6e, h). Consistent with the smallest delay in

out-growth of INH-resistant mutants, sub-inhibitory concentrations of BDQ had no detectable effect on the viability of an INH-resistant mutant compared to the drug-susceptible parent (Fig. 6f). TAC, which had the greatest effect on suppressing the growth of INH-resistant mutants in combination studies, was able to kill INH-resistant *katG* mutants at sub-inhibitory concentrations (Fig. 6g). This suggests that TAC, when used at sub-inhibitory concentrations, kills and selects against INH-resistant *katG* mutants, in contrast to BDQ, Q203 and PA824 that are only able to impair growth and delay the emergence of INH-resistant *katG* mutants.

To further investigate the targeting of collateral phenotypes to select against the emergence of drug-resistant strains, we determined the frequency of resistance against INH in the presence of sub-inhibitory concentrations of secondary collateral compounds, specifically (i) TAC (i.e. kills INH-resistant mutants at sub-inhibitory concentrations), and (ii) Q203 (i.e. impairs growth of INH-resistant mutants at sub-inhibitory concentrations). Consistent with 0.3x MIC Q203 only delaying the outgrowth of the INH-1 mutant, there was no change in the frequency of resistance against INH in the presence of 0.3× MIC of Q203 (Fig. 6i). TAC when used at 0.3x MIC alone had no effect on viability but was able to prevent the emergence of INH-resistant clones (Fig. 6i). Sub-inhibitory concentrations of TAC were also able to select against the emergence of INH resistance at increasing concentrations of INH. Specifically, there was a 2–3 log reduction in the frequency of resistance against INH at 1, 3 and 9× MIC in the presence of 0.3× MIC of TAC compared to INH alone (Fig. 6j). In conclusion, drug combinations that exploit collateral drug phenotypes can delay and, in some cases, select against the emergence of drug-resistant strains of *M. tuberculosis*.

## Discussion

There is an urgent need for not only new drugs, but rational approaches to the design of combination therapies that can shorten the length of treatment and prevent the emergence of drug resistance. Drug-resistant bacteria are frequently referred to as super-bugs because of their ability to resist clinically available antimicrobial treatments. While this has significant clinical implications, this overlooks the potential for mutations in core biological process also having consequences on downstream biological functions that make them more vulnerable to targeted inhibition. In this current study we have used a combination of antimicrobial susceptibility profiling, genomics, evolutionary studies, and ex vivo methods to demonstrate that collateral drug phenotypes are associated with drug resistance in *M. tuberculosis*, a leading cause of infectious disease mortality and morbidity.

Collateral resistance was frequently seen among antibiotics that shared similar mechanisms of action, such as QcrB inhibitors Q203 and TB47, MmpL3 inhibitors SQ109 and CPD1, and inhibitors of the HadABC dehydratase complex, TCL, and TAC. Unexpectedly, cross resistance was seen between CFZ resistant mutants and PA824. CFZ is a riminophenazine drug initially developed to treat TB, but was redirected for other uses due side effects[47]. CFZ has had a re-emergence of interest over the last ~10 years after several studies showed a benefit in adding CFZ in mouse models, as well as in human observational studies[48,49]. Prior work has suggested that by competing with menaquinone for reduction by type 2 NADH dehydrogenase (NDH-2) CFZ disrupts the NADH:NAD ratio and leads to increases in reactive oxygen species (ROS)[47,50]. However, recent work has cast doubt on the role of NDH-2 in CFZs mechanism of action[51]. PA824 has recently been approved by the FDA for use in multi-drug regimens[52]. The mechanism of action varies between replicating and non-replicating bacteria with processes involving cell wall damage and interruption of cellular respiration, respectively[53]. Although these drugs have distinct mechanisms of action, cross resistance between CFZ resistant mutants and PA824 was driven by mutations in *fbiC* and *fbiA*[36,54]. Genetic studies

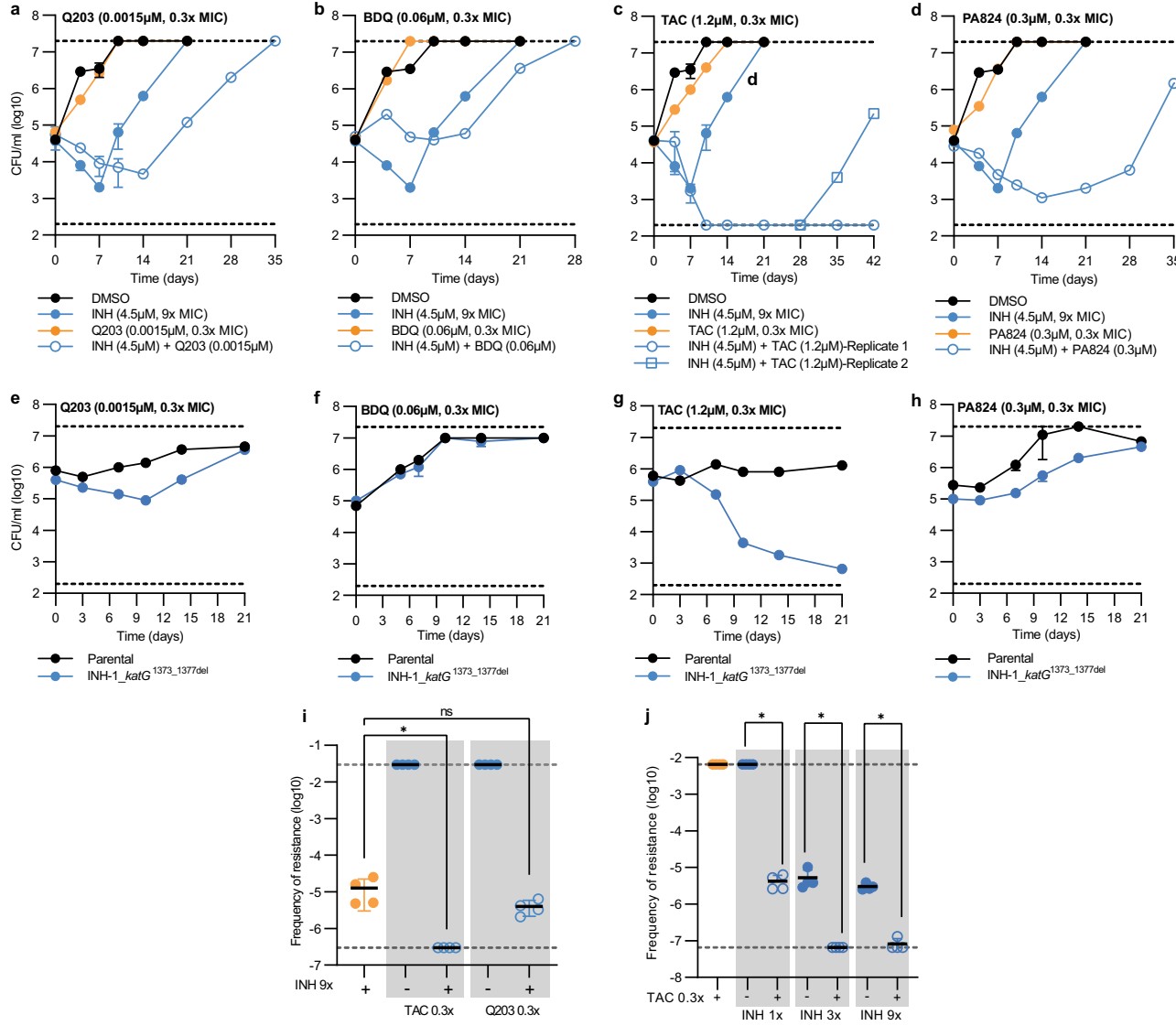

**Fig. 6 | Drug combinations that exploit collateral drug sensitivities delay the emergence of isoniazid resistance in vitro. a–d** Drug-susceptible *M. tuberculosis* was incubated with either INH at 9x the MIC, below MIC concentrations (i.e. 0.3× MIC of the drug-susceptible parent) of a second compound being either Q203, BDQ, TAC, PA824, or a combination of INH (9× MIC) and the second compound at concentrations below the MIC. **e–h** Time kill assays of selected INH-1 and drug-susceptible parent against sub-inhibitory concentrations (0.3× MIC). CFUs were determined on the stated days. **i, j** Frequency of resistance of drug-susceptible parent strain when challenged with antibiotics at various concentrations (single or in combination). For **i**, statistical significance was investigated using a one-way ANOVA, and corrected for multiple comparisons using the Dunnett's multiple comparisons test (95% CI). For **j**, statistical significance was investigated using a one-way ANOVA, and corrected for multiple comparisons using the Bonferroni multiple comparisons test (95% confidence interval). For **i, j**; ns = no significance, *$P ≤ 0.05$. For **a–h**, data are the mean ± range of biological duplicates from a representative experiment (*n* > 2 independent experiments). For **a–h**, the DMSO and INH (9× MIC) CFU plot is repeated across all five panels. For **c**, INH (9× MIC) + TAC (0.3× MIC) the biological duplicates are split into individual replicate plots as one replicate failed to evolve INH resistance. For **i, j**, data are the mean ± SD of four biological replicates from a representative experiment (*n* = 2 independent experiments). DMSO represents solvent control. Dashed lines represent upper and lower limits of detection.

validated this low-level increase in MIC (i.e. ~2-fold) against CFZ for PA824 resistant variants and expands on earlier studies which show that CFZ-resistant mutants typically have *rv0678* mutations. Our current study contradicts prior work demonstrating that the genetic deletion of F420 biosynthesis genes, rather than transcriptional inhibition as used in our studies, leads to increased sensitivity to CFZ[17,55,56]. Resolving these differences is essential, as this work highlights the potential for prior drug exposure (i.e. CFZ) to select for pre-existing resistance against other newly developed therapeutics (i.e. PA824).

Mutations in *rv0678*, a transcriptional regulator of the multi-substrate *mmpL5-mmpS5* efflux complex are associated with resistance to BDQ and CFZ[17,57]. Similar observations have been reported in *M. bovis*, *M. abscessus* and *M. smegmatis*[57–59]. Mutations in the TetR

repressor of *M. abscessus* also leads to concomitant upregulation of the *mmpL5-mmpS5* efflux pump, providing high-level resistance to TAC, an interaction not seen in this study[58]. In line with earlier findings, mutations in *rv0678* produced cross resistance between CFZ and BDQ[17]. Our results also support recent findings that mutations in *rv0678* provides resistance to PBTZ-169 (i.e. Macozinone[35,60]) in addition to extending the range of antibiotics to which mutations in *rv0678* provides low-level resistance against, including Q203. The observation of *rv0678* mutations in clinical populations, including in treatment naïve populations, further highlights the need to understand which pharmacophores are subject to efflux by *mmpL5-mmpS5* system as this may reduce the clinical lifespan and efficacy of compounds under clinical development[61]. A number of strains also had secondary

mutations in this study. While we hypothesize that the majority of secondary mutations don't directly contribute to collateral phenotypes, further investigation is needed to verify this.

INH is an integral component of first line treatments for drug-susceptible tuberculosis. Unfortunately, there is widespread INH resistance, with aggregated data from 156 countries or territories during 2003–2017 estimating the overall prevalence of INH resistance at 8.4% among all TB patients[62]. Resistance to INH is most commonly a result of mutations in the *katG* catalase-peroxidase that prevents activation of the INH prodrug, with approximately 78.6% of INH-resistant isolates having a mutation in *katG*[62]. Additional INH resistance mutations are found in InhA, the primary target of INH, and other genes including *ndh* at a lower frequency[23]. Characterizing the consequences of *katG* mutations in INH-resistant strains mutants could identify collateral drug sensitivities that could be therapeutically exploited. Here, we have discovered several examples of collateral drug sensitivities in laboratory-evolved INH resistant isolates. First, INH resistant mutants exhibited increased sensitivity to inhibition by BDQ, PA824, TB47, Q203 and TAC. Inactivation of the *katG* catalase-peroxidase also altered the killing dynamics of these collateral compounds under both in vitro and ex vivo conditions, with all drugs impacting the INH resistant mutant to a much greater extent. Importantly, these collateral drug phenotypes could be exploited in drug combinations to alter the emergence of INH-resistant strains. TAC targets key components of mycolic acid synthesis, interrupting cell wall formation and while it has been historically used in the treatment of TB, it has been largely discontinued due to toxicity issues[34]. Our findings suggest that TAC or potentially an alternative inhibitor of mycolic acid synthesis can actively select against INH-resistant mutants.

Therapeutic targeting of collateral sensitivities has the potential to increase treatment efficacy and suppress the development of antibiotic resistance[63,64]. Despite these promises valid concerns remain regarding the conservation of collateral phenotypes across different genetic backgrounds and under different conditions[65]. Among geographic lineages of *M. tuberculosis* differences in genetic requirements can influence antibiotic susceptibility and evolvability of drug resistance[66–68]. Furthermore, compensatory mutations and combinations of resistance genotypes have been shown to alter the fitness costs of mono and multi-drug-resistant strains[69]. While collateral networks have been reported as being conserved among clinical isolates of *E. coli*[8], determining whether these collateral pathways are conserved among diverse geographical lineages and multi-drug-resistant backgrounds of *M. tuberculosis* requires further investigation. Importantly, this current study goes some way towards addressing these issues as we observed instances of collateral drug phenotypes being conserved among genetically diverse drug-resistant strains. For example a *katG* (i.e. INH), *atpE* (i.e. BDQ) and multiple *rpoB* (i.e. RIF and FIX) mutants all had increased sensitivity to growth inhibition by PA824 and numerous strains had increased sensitivity to the cytochrome $bc_1:aa_3$ inhibitor Q203. While the collateral drug sensitivity for the INH and BDQ resistant mutant against PA824 could be exploited within infected macrophages, the RIF resistant was more tolerant of killing by PA824 highlighting the need to determine how experimental conditions and metabolic state may alter the observed collateral sensitivities[65]. Furthermore, the collateral sensitivity to PA824, did not alter the ability of *katG*, *atpE* and *rpoB* mutants to evolve secondary drug resistance. Whether this is the case for all collateral phenotypes, and whether the acquisition of multiple resistance mutations effects the stability of collateral phenotypes requires further investigation[70].

Drug cycling, in which drugs are switched throughout treatment to counter the evolution of resistance, is one strategy for exploiting on collateral phenotypes and providing a personalized approach to infectious disease therapy. Mathematical models predicting the efficacy of drug cycling in a 'laboratory' and 'patient' model have highlighted the potential benefits of drug cycling[63]. Comparable to prior drug-cycling experiments on other pathogens our results demonstrate that collateral

sensitivities can be targeted to select against drug-resistant strains of *M. tuberculosis* in both a homogenous and heterogenous population[12,71]. Increasing advancements in *M. tuberculosis* diagnostics allows for resistance genotypes to be rapidly determined throughout the course of infection[72]. Combined with drug resistance in *M. tuberculosis* being restricted to spontaneous chromosomal mutations, rather than horizontal gene transfer, and *M. tuberculosis* having a relatively low mutation rate in vitro and in vivo, *M. tuberculosis* could be an interesting candidate for drug-cycling regimens[73,74]. To achieve this, there is a need to understand how drug cycling and prior drug resistance influences the evolution of secondary drug phenotypes in *M. tuberculosis*. Furthermore, whether drug cycling could be applied on a large scale and across a diverse array of health care settings as would be needed for the treatment of *M. tuberculosis* requires consideration.

In this current study, we provide evidence that drug-resistance in *M. tuberculosis* is associated with collateral drug phenotypes and that drug combinations targeting these phenotypes can prevent or suppress the evolution of drug resistance. Additional work extending these phenotypes to alternative conditions, relevant animal models, pre-existing drug-resistant strains, diverse geographic lineages, and potential compensatory mutations is crucial for realizing the therapeutic potential of this work. Despite this, this work in combination with the contributions of others highlights how an improved understanding of the biology and consequences of drug-resistance can identify unique therapeutic strategies to revolutionize the fight against anti-microbial resistance.

## Methods
### Bacterial strains and growth conditions
The avirulent strain of *M. tuberculosis* used in this study, mc²6206 (Δ*leuCD*, Δ*panCD*) is approved for use under PC2 conditions at The University of Otago. All derivatives of mc²6206 used in this study are listed in Table 2. All mc²6206 strains (parental and resistant derivatives) were grown and maintained in 7H9 liquid media or on 7H11 solid media (Becton Dickinson, #271310 and #212203) supplemented with OADC (0.005% oleic acid [Sigma, #O1008], 0.5% bovine serum albumin [pH Scientific, #PH100], 0.2% dextrose [Sigma, #G8270], 0.085% catalase [Sigma, #02071]), 0.05% tyloxapol (Sigma, #T8761), 50 μg/mL leucine (Sigma, #L8000) and 25 μg/mL pantothenic acid (Sigma, #21210) and incubated at 37 °C. Strains were maintained in 10 mL 7H9 liquid culture in 30 mL square inkwell bottle (Nalgene, #2019-0030) or T25 culture flasks (Sigma, #156367). Antibiotics used in this study, their target and catalogue number are provided in Supplementary Table 1.

### Resistant mutant isolation
To isolate drug-resistant strains of *M. tuberculosis*, we utilized an experimental approach consistent with prior studies investigating in vitro drug resistance in *M. tuberculosis*[67,75,76]. While alternative methods have been reported, the long doubling time of *M. tuberculosis* makes progressive exposure to increasing concentrations of antibiotics experimentally prohibitive when utilizing multiple antibiotics. Importantly, comparative studies in *E. coli* have demonstrated that key adaptions are conserved across adaptative evolutionary protocols[77]. Briefly, the mc²6206 parental strain was grown in 10 mL 7H9-supplemented media from a starting CFU of $5 \times 10^4$/mL (i.e. $OD_{600}$ 0.00005) until they reached an $OD_{600}$ of ~0.4–1. This was repeated with 12 independent cultures. All cultures were harvested by centrifugation at $3220 \times g$ for 10 min and resuspended in fresh media at a combined volume of 3 mL. Culture was further diluted along a 3-point, 10-fold gradient in 7H9-supplemented media. 100 μL of each dilution was inoculated onto 7H11-supplemented agar plates containing either 3, 10 or 30× the minimum inhibitory concentration (MIC) of the chosen compound. Plates were incubated at 37 °C until growth of individual colonies was visible. To confirm a resistance phenotype, isolated colonies that grew in the presence of compounds were re-streaked

**Table 2 | Drug-resistant *M. tuberculosis* mc²6206 strains generated and used in this study**

| Resistant mutant | Isolating antibiotic | Isolating concentration (µM) |
|---|---|---|
| RIF-1 | Rifampicin | 0.12 |
| RIF-11 | | 0.4 |
| RIF-21 | | 1.2 |
| FIX-1 | Fidaxomicin | 30 |
| LEV-1 | Levofloxacin | 1.2 |
| LEV-2 | | 1.2 |
| LEV-11 | | 4 |
| LZD-1 | Linezolid | 4.5 |
| LZD-2 | | 15 |
| LZD-3 | | 4.5 |
| KAN-1 | Kanamycin | 6 |
| KAN-32 | | 60 |
| STRP-2 | Streptomycin | 0.18 |
| STRP-8 | | 0.6 |
| STRP-20 | | 1.8 |
| CAP-1 | Capreomycin | 3 |
| CAP-10 | | 10 |
| CAP-12 | | 10 |
| PA824-8 | Pretomanid | 1.5 |
| PA824-21 | | 5 |
| PA824-41 | | 15 |
| CFZ-1 | Clofazimine | 0.18 |
| CFZ-3 | | 0.18 |
| CFZ-6 | | 0.18 |
| BDQ-2 | Bedaquiline | 2.7 |
| BDQ-14 | | 2.7 |
| TB47-42 | TB47 | 0.15 |
| Q203-1 | Q203 | 0.06 |
| Q203-13 | | 0.2 |
| Q203-18 | | 0.6 |
| AZ7-11 | AZ7371 | 24 |
| PBTZ-1 | PBTZ-169 | 0.0015 |
| PBTZ-2 | | 0.0015 |
| INH-1 | Isoniazid | 0.3 |
| INH-11 | | 1 |
| INH-21 | | 3 |
| ETH-1 | Ethionamide | 18 |
| ETH-2 | | 18 |
| ETH-6 | | 18 |
| CPD-1-1 | CPD-1 | 1.5 |
| CPD-1-3 | | 1.5 |
| CPD-1-4 | | 1.5 |
| THP-18 | Thiophene-2 | 3 |
| THP-23 | | 10 |
| TAC-1 | Thioacetazone | 3 |
| TAC-16 | | 10 |
| TAC-9 | | 30 |
| TCL-1 | Thiocarlide | 50 |

onto 7H11-supplemented agar plates containing the same compound at the same concentration. A drug-susceptible (DS) parental strain was also streaked as a negative control. Confirmed resistant mutants were transferred into 10 mL 7H9-supplemented media and frozen as glycerol stocks for future use.

**Liquid media minimum inhibitory concentration (MIC) assays**

MIC assays were performed as previously described, with modifications[39,78,79]. Briefly, inner wells (rows B–G, columns 2–11) of a 96-well flat-bottomed microtiter plate (ThermoFisher Scientific) were filled with 75 µL 7H9-supplemented media. Outer wells were filled with 150 µL 7H9-supplemented media and left as media only controls. Compound plates were independently prepared in a 96-well U-bottom microtiter plate by adding the highest concentration of compound to column 2 and diluted 3-fold along a 9-point dilution in appropriate solvents. Column 11 was kept as solvent only. 1.5 µL from each well of the compound plate was then dispensed into a corresponding well of the assay plate. DMSO was kept to a final concentration of 1% in assay plates. All mc²6206 strains (parental and resistant derivatives) were diluted to an $OD_{600}$ of 0.01. Seventy-five µL of diluted culture was added to inner wells of the 96-well flat-bottomed microtiter plate containing compound to achieve a starting $OD_{600}$ of 0.005 in a final volume of 150 µL. Plates were incubated at 37 °C for 10 days without shaking. After 10 days, plates were covered with plate seals, shaken for 1 min and the $OD_{600}$ was determined using a Varioskan Flash microplate reader (ThermoFisher Scientific). MICs of each compound were determined from dose response curves as previously described[39,78,80]. Briefly, the $OD_{600}$ reads of the outer wells were averaged to provide a background value that was removed from the raw $OD_{600}$ reads of all inner wells. The $OD_{600}$ values of columns 2–10 were represented as relative to the solvent only column (column 11). The MIC was determined using a nonlinear fitting of data to the Gompertz equation[80].

CRISPRi MICs were performed as above with the following variations. Knockdown strains were pre-depleted for 5 days in the presence of 300 ng/mL anhydrotetracycline (ATc) and 20 µg/mL kanamycin prior to inoculation into MIC assays. 300 ng/mL ATc was included throughout MIC assays to ensure gene repression. CFZ and PA824 were diluted twofold along a 9-point dilution in appropriate solvents. Data were interpreted as above.

**Collateral susceptibility profile determination**

The susceptibility profiles of three resistant mutants isolated per compound were determined for all 23 compounds used in this study. Where possible, these three resistant mutants comprised all concentrations used for isolation, i.e. used a single mutant isolated at 3, 10 and 30× MIC. When resistant mutants were not isolated against higher concentrations (i.e. 30× MIC), additional mutants from lower concentrations were used. All liquid MIC assays were performed as described above. Each MIC replicate assay included the mc²6206 parental strain to be used as a reference as well as selected resistant mutants. Fold changes in susceptibility of each resistant mutant against each compound were determined relative to the parental strains used in each replicate experiment. For each resistant strain, all MICs and fold changes in susceptibility relative to the parental strain were determined from at least four biological replicates. Collateral susceptibility was defined by an average threefold change in MIC (decrease = sensitization, increase = resistance) across replicate experiments. Heatmaps showing changes in susceptibility were visualized using GraphPad-Prism.

**Minimum bactericidal concentration (MBC)**

To determine the minimum bactericidal activity of compounds, 96-well plates were prepared as previously described for the liquid MIC assays. To determine CFU counts, 150 µL of each well containing compound and culture was transferred to a new 96-well plate after 10 days incubation. The transferred samples were diluted along a 3-point, 10-fold dilution curve in 7H9 with pantothenic acid and leucine. 5 µL of each dilution was spotted onto 7H11 supplemented with OADC, leucine and pantothenate acid. Plates were incubated at 37 °C for 4–5 weeks until growth was visible, and colonies could be counted. To

determine starting CFU count, culture diluted to an $OD_{600}$ of 0.01 on day 0 was diluted and spotted as described above.

## Construction of CRISPRi knockdown strains

To achieve transcriptional knockdowns in *M. tuberculosis*, a 20–25 bp target sequence downstream of a PAM sequence was identified in each desired gene. The 20–25 bp sequence from the template and non-template strand were ordered as oligos with 5' GGGA and AAAC overhangs suited for introduction to the CRISPRi plasmid (i.e. pJLR965[38]). Information regarding each individual oligo and sgRNA (i.e. PAM sequence, predicted PAM strength and target sequence) are provided in Supplementary Table 2. Oligos were annealed and cloned into pJLR965 using BsmB1 restriction enzyme (NEB #R0580). The plasmids were sequenced verified and transformed into *M. tuberculosis* strain mc²6206 using previously published methods[39]. Knockdown strains were grown in media with 20 μg/mL kanamycin to maintain the plasmid.

## Killing kinetics

To determine killing kinetics, cultures of mc²6206 strains were diluted to an $OD_{600}$ of 0.1. 500 μL of $OD_{600}$ adjusted culture was added to 9.5 mL 7H9-supplemented media to achieve a starting $OD_{600}$ of 0.005 in a 30 mL inkwell or T25 flask. 50 μL of diluted compounds were added, with DMSO at a final concentration of 0.5%. In co-treatment experiments, antibiotics were added so the final concentration of DMSO was ≤1%. Culture was removed on stated days to measure $OD_{600}$ as well as diluted and spotted as described above for MBC to determine the number of viable colonies.

## Co-culture time kill experiments

To investigate if collateral sensitivities could be exploited in a heterogenous population, competition assays were performed using previously published protocols[76,81]. Briefly, strains were diluted to an $OD_{600}$ of 0.1 and combined at a 1:1 ratio, with 0.5 mL of the 1:1 mixture being added to 9.5 mL 7H9-supplemented media in a 30 mL inkwell. 50 μL of diluted compounds were added to appropriate inkwells, with DMSO at a final concentration of 0.5%. Culture was taken on the stated days and spotted onto 7H11 media with or without 9x MIC INH. To determine the proportion of parental strain in the population, colony counts from the 7H11 + 9× INH were removed from the 7H11 only plate ($CFU^{7H11}$–$CFU^{7H11+9× INH}$). Counts from the 7H11 with INH were used to quantify the proportion of the population that was INH resistant.

## Frequency of resistance studies

To determine the frequency of resistance for mc²6206 parental strain and resistant derivatives, frequency of resistance studies were performed following previously published protocols[67,75]. Briefly, strains were inoculated in replicates of four at a starting OD of 0.005 in 10 mL of 7H9-supplemented media and left to grow until an $OD_{600}$ of 0.4–1.0. Supplemented 7H11 agar was prepared with appropriate antibiotics or DMSO controls and poured as 6mL volumes into compartments of a ¼ sectioned round plates (NEST Biotechnologies, CAT #752031). Replicate growth cultures were individually harvested by centrifugation at 3220 × *g* for 10 min and resuspended in 2 mL supplemented 7H9. Strains were serially diluted along a 7-point 10-fold dilution curve. 20 μL of diluted culture was spotted and spread. DMSO control plates were spotted down to $10^{-7}$ dilution to determine the number of countable colonies, while antibiotic containing plates were spotted down to $10^{-3}$ dilution. Plates were incubated at 37 °C for 4 weeks, until growth was visible, and colonies counted.

## THP-1 macrophage infection studies

The antibiotic sensitivity of *M. tuberculosis* within THP-1 infected macrophages were determined using previously described protocols[82], with modifications. Briefly, the human monocytic cell line

THP-1 (ATCC Cat# TIB-202) was cultured in standard RPMI 1640 macrophage medium supplemented with 10% inactivated fetal bovine serum and 1 mM sodium pyruvate at 37 °C with 5% $CO_2$. THP-1 monocytes ($5 × 10^5$ cells/well) were differentiated overnight using 100 ng/mL phorbol myristate acetate (PMA) and seeded in a 24 well-plate. The next day differentiated macrophages were infected with a mid-logarithmic phase culture of *M. tuberculosis* mc²6206 parent strains or resistant derivatives (OD 0.4–0.8) at a multiplicity of infection (MOI) of 10:1 (10 bacteria/1 cell). Infection was allowed to proceed for 1 h. Cells were then washed 3 times with pre-warmed complete RPMI to remove extracellular bacilli. RPMI media containing supplements (pantothenic acid 25 μg/mL and leucine 50 μg/mL), 0.1% BSA and compounds at varying concentrations were added to the infected cells and incubated at 37 °C with 5% $CO_2$. After 3 days, infected cells were lysed in distilled water containing 0.1% tyloxapol for 5 min at room temperature to determine the number of CFU/mL on Middlebrook 7H11 OADC agar supplemented with (pantothenic acid 25 μg/mL and leucine 50 μg/mL). The percentage of *M. tuberculosis* (mc²6206) cell viability was determined by normalizing CFU/mL counts at day 3 following compound treatment relative to the DMSO control.

## DNA isolation and whole-genome sequencing

DNA was isolated from cultures of *M. tuberculosis* mc²6206 strains (parental and resistant derivatives) using previously described protocols[66]. Briefly, 10 mL of culture grown in 7H9-supplemented media was harvested by centrifugation at 3220 × *g* for 10 min. The pellet was resuspended in 0.5 mL TE-buffer and the culture inactivated by adding an equal volume chloroform:methanol (2:1) and mixed by inverting for 5 min at room temperature. The mixture was then centrifuged at 3220 × *g* for 10 min, the supernatant removed, and pellets allowed to air dry. Pellets were resuspended in 0.5 mL TE-buffer, to which 50 μL of lysozyme (10 mg/mL in 1 M Tris, pH 8.0) was added and incubated at 37 °C overnight. 55 μL of 10% SDS and 6 μL proteinase K (20 mg/mL, Thermo-Fisher: EO0492) were added and incubated at 65 °C for 30 min. An equal volume of phenol-chloroform (1:1) was added and left at room temp for 30 min with occasional mixing. The entire mixture was transferred to a Maxtract High Density tube (Qiagen: 129056) and centrifuged following manufacturers guidelines to separate phases. The upper aqueous phase (approx. 500 μL) was transferred to a new Maxtract tube to which 1.25 μL of RNase A (10 mg/mL, Thermo-Fisher: EN0531) was added at a final concentration of 25 μg/mL and incubated for 30 min at 37 °C. An equal volume of phenol-chloroform (1:1) was added, centrifuged and upper aqueous phase was again transferred to a new Maxtract tube. An equal volume of chloroform was added, inverted, and centrifuged. The upper aqueous phase was transferred to a new Eppendorf tube, to which 0.1 volume of 3 M sodium acetate pH 5.2 and 1 volume of isopropanol was added, mixed and incubated at −20 °C for 1 hr. Tubes were centrifuged at ≫16.2 K × *g* for 30 min at 4 °C. The supernatant removed and replaced with 0.5 mL of 70% ethanol to the resulting pellet. Tubes were centrifuged for 5 min, with supernatant removed and pellets allowed to air dry. Once dried, pellets were resuspended in 70 μL TE buffer, concentration measured and integrity of isolated gDNA determined by running samples on a 1% agarose gel.

Whole-genome sequencing (Illumina 150 bp paired end reads) was performed at The Microbial Genome Sequencing Centre (Pittsburgh, USA). Resulting reads were paired and trimmed in Geneious Prime® 2020.2.4 using the BBDuk[83] plugin. The *M. tuberculosis* mc²6206 parental strain was mapped to the H37Rv genome (Genbank accession number: NC_000962) using Geneious Bowtie-2[84] plugin to generate a consensus sequence. All subsequent *M. tuberculosis* mc²6206 resistant strains were mapped to the mc²6206 parental strain consensus sequence using Geneious Bowtie-2 plugin. SNPs and genomic variants were identified using the Geneious 'Find Variation/SNPs' built-in tool. Variants were called with a minimum coverage of 20, and

a minimum variant frequency of 80%. All mutations identified in each drug-resistant strain and the variant frequency are described in Supplementary Dataset 2. All raw Illumina reads are available at the NCBI-SRA (accession number: PRJNA914416).

**Reporting summary**

Further information on research design is available in the Nature Portfolio Reporting Summary linked to this article.

## Data availability

The source data used to generate the figures in this study are available through figshare (https://figshare.com/articles/dataset/Waller_et_al_2023_Collateral_TB_Source_Data_xlsx/22152056). All raw Illumina reads used to generate whole-genome sequences are available at NCBI-SRA (accession number: PRJNA914416). Additional data are available from the corresponding author upon request.

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

## Acknowledgements

This research was financially supported by the Maurice Wilkins Centre for Molecular Biodiscovery and the Marsden Fund (Royal Society of New Zealand) (grant number UOO1807) awarded to M.B.M. N.J.E.W. was supported by a University of Otago Doctoral Scholarship. We thank all members of the Cook lab for helpful discussions. We thank Kevin Pethe for providing Q203, Xiaoyun Lu for providing TB47 and Tanya Parish for providing CPD1.

## Author contributions

G.M.C. and M.B.M. conceptualized the study. N.J.E.W., C.-Y.C. and M.B.M. designed, performed and analysed the experiments. N.J.E.W. and M.B.M. wrote the manuscript with input from other authors.

## Competing interests

The authors declare no competing interests.
