## [Peer review file · Nature Communications]

REVIEWER COMMENTS

Reviewer #1 (Remarks to the Author):

First of all, I have to note that I am not an expert in lab procedures but in bioinformatics and genomics. So, the fact that I have not written any comment regarding this aspect does not mean that lab procedures and techniques are ok but that I don't have the expertise to evaluate it properly.

In this article, Waller et al. described how mutant strains resistant to specific drugs have either enhanced resistance or susceptibility to other drugs, mediated by the variants responsible for the primary resistance. They generated a library of drug-resistant (DR) isogenic mutants from an attenuated *M. tuberculosis* strain. Later, they tested the enhanced susceptibility or resistance to drugs different to the one used to evolve each of the mutants, in comparison with the parental strain. Furthermore, they demonstrated that the patterns observed worked not only in-vitro but also when infecting human macrophages. Finally, they tested that different combinations of drugs that create collateral enhanced susceptibilities can be used together to delay the apparition of drug-resistant mutants in-vitro.

I would like to congratulate the authors for such a well done job. There is a huge amount of experimental and analytical work performed. The manuscript is well written, figures are illustrative and, to my knowledge, methods seem to be clearly explained. I have no comments to improve the job made by the authors regarding my expertise, but I think that there are few questions that we can discuss about:

-Which could be the genomic determinants of such collateral phenotypes? The authors discuss in the 'Collateral drug sensitivity associated with drug resistance in *M. tuberculosis*' section that some drug-resistant mutations for which they have previous information seemed to be related with the phenotypes while some others not (ie. strains with the same DR mutation but having different sensibilities enhanced). BUT if there are cases in which secondary mutations seem to be responsible for the observed phenotype (when the same DR mutation shows different sensitivities) this may also happen when the same mutation shows the same sensitivities. We can not preclude that the observed mutation is responsible for the observed trait unless specifically testing it, constructing a mutant with only this specific mutation and testing the phenotypes again.

-It would be nice if authors could include, as supp material, the total number of annotated mutations called in every evolved strain. It will allow other researchers to inspect data to look for potential secondary mutations (or compensatory ones) that may be also involved in the phenotypes observed.

-Based on the data generated by the authors... Is it possible to improve current standard and MDR treatments in order to diminish the likelihood of DR emergence while maintaining treatment effectiveness?

-Regarding the Methods section, please detail a bit more how the variant calling step was performed after mapping, at the end of the "DNA isolation and WGS" section.

Reviewer #2 (Remarks to the Author):

This manuscript by Waller et al aims to study antibiotic cross-resistance (CR) and collateral sensitivity (CS) in *M. tuberculosis*. The phenomenon of CS, which has previously been examined in several studies going back 15 years or so, involves cases where the evolution of antibiotic resistance can result in increased sensitivity to unrelated drugs. A majority of the previous studies have examined this phenomenon in *E. coli* and *P. aeruginosa*. In this manuscript, the authors have performed a thorough study of CS (and CR) for 23 different drugs using an avirulent variant of *M. tb*. Using selection experiments combined with genetic and phenotypic tests they show in vitro and in macrophages that CS can eliminate/reduce the emergence of resistant mutants in the population. Overall, I think this is a well-designed and interesting study and the use of *M. tb* as the study organism, a major pathogen where resistance emerges exclusively by mutation, further increases its potential clinical value.

1. A key question regarding the potential clinical application of CS is how conserved it is between different isolates of *M. tb*? Even though the study was not designed to examine this it would be good if the authors would explicitly discuss this in the context of other species (e.g. *E. coli*) and the likelihood of this being a concern for *M. tb*. Personally, I think lack of conservation will be a major obstacle in utilizing CS clinically but perhaps *M. tb* is different because of its small pan-genome and limited genetic variability?

2. One weakness of all (?) studies of CS is that the conditions used to show that CS can reduce resistance evolution are often chosen to poise the system such that the expected effect can be seen. Thus, mutation supply rates (i.e. population size), drug level, time of exposure etc are typically not systematically varied to determine how these variables will affect the extent of CS. It would be useful if the authors discussed why the particular conditions were chosen and how altering them might change the ability of CS to reduce resistance evolution.

3. It would be useful if the authors state in the first section of the results (in text or supp table) for all the different clinically used drugs how representative the isolated in vitro mutants are (i.e. are they the

mutations that are typically found in clinical isolates). Furthermore, state in text/table/fig (could for example be added in parenthesis next to the resistance mutations in fig 1) whether other mutations than the expected resistance mutations were present in the strains.

4. Fig. 1 is misleading with regard to the colours chosen to illustrate the extent of CR or CS. Since no cases of 9x or 27x MIC-fold changes were observed for CS I would remove them from the X-axis. This will also clearer emphasize that the big effects are very much skewed towards cross-resistance rather than collateral sensitivity.

Reviewer #3 (Remarks to the Author):

Waller et al. have investigated the development of collateral sensitivity effects in *M. tuberculosis*. They find multiple examples of collateral effects and show how these can be exploited to counter-select resistant isolates. This is an interesting and thorough piece of work (especially considering that it was performed in *M. tuberculosis*) that will be a worthwhile addition to the growing body of knowledge of drug resistance development in MTB. However, I do have some issues concerning the strong statements regarding the clinical potential of using collateral sensitive in a clinical setting. I think additional experiments are needed here and the conclusions should be toned down.

Major comments:

1. In the sections concerning treatment strategies (203-222 and 308-371), the authors choose sub-MIC antibiotic concentrations. In line 43-45 the authors write that ‘... patient non-compliance and suboptimal drug penetration frequently results in treatment failure, driving the evolution and spread of drug resistance...’. Keeping this in mind, is it (i) possible to achieve specific sub-MIC concentrations within the human body, and (ii) would this not result in even more resistance development (i.e. see <https://doi.org/10.1038/s41467-018-04059-1>).

2. Lines 188-202: The authors call these secondary mutations which might suggest that they appeared during the drug selection (next to the primary resistance mutations). From table S2 it is obvious that this is not the case. The identified mutations show that the strain stock has undergone an extensive multi-step adaptive evolution process (for example to adapt to the media or the auxotrophy mutations). I suggest that the authors (i) make a clear comment about this in the manuscript, and (ii) add a supplementary figure with the evolutionary trajectories of the clearly identifiable background mutations and mark out where in the tree the resistant mutations can be found. This will help to make the

distinction where discrepancies in the collateral sensitivity profile are caused by the strain background and where due to the drug selection. Also, I suggest to isolate a new 'wild-type' strain that is adapted to media/auxotrophies for future work to reduce the diversity in the strain background.

3. Lines 270-273 and 362-365: If the collateral sensitivity phenotype causes no change in the development of new resistance, then where is the benefit of combining two drugs with collateral effects compared to two drugs without? In a clinical setting it is not feasible (or smart) to supply specific sub-Mic concentrations of antibiotics. The question that should be answered is if combining two drugs with a collateral sensitivity profile are better in preventing resistance emergence than two drugs without sensitivity. I don't think the current experiments actually show if this is the case.

Minor comments:

4. According to reference 1, the first-line treatment for MTB includes pyrazinamide and ethambutol. It seems like pyrazinamide was not included in the study and resistance mutants for ethambutol were removed due to slow growth. This is unfortunate since it eliminates half of the first-line drugs out of the study. It would be important to know if the current treatment regimen already uses collateral sensitivity effects or not.

5. Lines 106-110: I don't see an example of more than three mutants for a single drug in figure 1 so I would drop the 'at least' from the sentence.

6. Lines 110-119: If a 3-fold dilution series was prepared, then how do you measure a 2-fold change? I suggest to remove the 2-fold changes from the results since they should not be considered significant. Having the additional shade of blue in figure 1 makes it harder to see what is actually significant.

7. Figure 1: The shades are too similar, especially if the non-significant 2-fold changes are included. I can honestly not tell if any of the drugs have 9-fold reduction in MIC or not. In the same direction, a 27-fold change is given a color but it doesn't seem to exist in the data set.

8. How do the isolated mutations compare to the once found in clinical MTB isolates?

9. For rifampicin only a single *rpoB* mutation was isolated (*rpoB* S450L). This is the most common clinical mutation in MTB but there are many others that are clinically relevant. Do these also show collateral sensitivity effects? Maybe it would be possible to isolate at least a couple of the other relevant

mutations and test for sensitivity to PA824. Also rpoB S450L often with secondary compensatory mutations in rpoA, rpoB, or rpoC. Could that effect the collateral sensitivity profile?

Reviewer comments in bolded text.

Author responses in normal text.

Reviewer #1 (Remarks to the Author):

First of all, I have to note that I am not an expert in lab procedures but in bioinformatics and genomics. So, the fact that I have not written any comment regarding this aspect does not mean that lab procedures and techniques are ok but that I don't have the expertise to evaluate it properly.

In this article, Waller et al. described how mutant strains resistant to specific drugs have either enhanced resistance or susceptibility to other drugs, mediated by the variants responsible for the primary resistance. They generated a library of drug-resistant (DR) isogenic mutants from an attenuated *M. tuberculosis* strain. Later, they tested the enhanced susceptibility or resistance to drugs different to the one used to evolve each of the mutants, in comparison with the parental strain. Furthermore, they demonstrated that the patterns observed worked not only in-vitro but also when infecting human macrophages. Finally, they tested that different combinations of drugs that create collateral enhanced susceptibilities can be used together to delay the apparition of drug-resistant mutants in-vitro.

I would like to congratulate the authors for such a well done job. There is a huge amount of experimental and analytical work performed. The manuscript is well written, figures are illustrative and, to my knowledge, methods seem to be clearly explained. I have no comments to improve the job made by the authors regarding my expertise, but I think that there are few questions that we can discuss about:

Comment 1: Which could be the genomic determinants of such collateral phenotypes? The authors discuss in the 'Collateral drug sensitivity associated with drug resistance in *M. tuberculosis*' section that some drug-resistant mutations for which they have previous information seemed to be related with the phenotypes while some others not (ie. strains with the same DR mutation but having different sensibilities enhanced). BUT if there are cases in which secondary mutations seem to be responsible for the observed phenotype (when the same DR mutation shows different sensitivities) this may also happen when the same mutation shows the same sensitivities. We can not preclude that the observed mutation is responsible for the observed trait unless specifically testing it, constructing a mutant with only this specific mutation and testing the phenotypes again.

Response 1: Whilst we agree with the reviewer that secondary mutations are present in drug-resistant strains. In our study, secondary mutations were predominately conserved among resistant mutants when they were isolated from the same culture that was used in resistant mutant isolation studies (see Table S2). These resistant mutants that have shared secondary mutations each have distinct resistance/sensitivity profiles.

For example, INH-21 which has an increased sensitivity to PA824, has one secondary mutation, *ppsB* (see supplementary Table 2). This mutation is also present in several other mutants including KAN-1, CAP-10, and PBTZ-1 that do not have increased sensitivity to PA824. Furthermore, RIF-1, BDQ-2 and INH-1 all share an increased sensitivity to PA824, yet do not share any secondary mutations. For these reasons we hypothesized that the majority of secondary mutations would not influence the collateral phenotypes we observed.

We acknowledge that with our experimental design, there is the possibility that some secondary mutations may be responsible for the collateral phenotypes of strains with the same drug-resistance mutation. This was addressed in our initial submission when there was a

disagreement between strains with the same DR-mutation (e.g., THP-resistant mutants). To highlight these limitations, we have added the following statements to our revised manuscript

See Results section: “Collateral drug sensitivity associated with drug resistance in *M. tuberculosis*” (Pages 10/11, lines 205-215)

“Drug-resistant strains isolated in this study had a unique collection of secondary mutations (Table S2). Many secondary mutations were conserved among resistant mutants each with separate resistance/sensitivity profiles that were isolated from the same culture that was used in resistant mutant isolation (Table S2). Consequently, we hypothesized that the majority of secondary mutations would not influence the collateral phenotypes we observed. For example, KAN-1, PBTZ-1, and INH-21 all harbour the same point mutation in *ppsB* (I896N), a gene proposed to be involved in the biosynthesis of phthiocerol dimycocerosate (PDIM)⁴². However, KAN-1 does not have increased sensitivity to any other antibiotic, and PBTZ-1 only shares two of the five increased sensitivities experienced by INH-21. INH-1 and INH-11 that have comparable collateral profiles to INH-21 had no mutations in *ppsB*. In some instances, secondary mutations did influence collateral phenotypes as resistant mutants with identical primary drug-resistance mutations had different responses to drug challenge (Figure 1, Table S2). For example, two strains with resistance to the Pks13 inhibitor thiophene-2 (THP)⁴³, (i.e. THP-18 and THP-23) both harbored the same point mutation (*pks13*^{T427A}), yet had different sensitivities to several antibiotics. THP-18 harbored a mutation in *YrbE2A*, a predicted ABC transporter permease, that was not observed in THP-23, or any other mutant isolated used in this study. Consistent with the THP-18 collateral sensitivities, chemical genetic studies have demonstrated the loss of *YrbE2a* increased sensitivity to RIF and BDQ⁴⁴, (Figure 1 and S3C-D). THP-18 also had a low-level sensitivity (i.e., 2-3-fold) to other antibiotics including Q203, PA824 and PBTZ-169 (Figure 1).

See Discussion section. (Pages 20/21, lines 460-463):

“A number of strains also had secondary mutations in this study. Whilst we hypothesize that the majority of secondary mutations don’t directly contribute to collateral phenotypes, further investigation is needed to verify this.”

Comment 2: It would be nice if authors could include, as supp material, the total number of annotated mutations called in every evolved strain. It will allow other researchers to inspect data to look for potential secondary mutations (or compensatory ones) that may be also involved in the phenotypes observed.

Response 2: We agree with the reviewer that this additional information would be useful to other researchers. We have updated Table S2 in the revised manuscript to include the following:

- The total number of mutations present in each strain when aligned with the parental strain.
- The experiment from which each mutant was isolated to highlight a shared parent culture.
- For the primary-DR mutations we have provided information as to whether that mutation is observed clinically.
- For each identified mutation we have provided the variant frequency (i.e., the proportion of reads that had that mutation).
- In line with the Nature publishing requirements all raw reads used to construct each of the DR-mutant genomes have been made available in NCBI-SRA (project number: PRJNA914416).

Comment 3: Based on the data generated by the authors... Is it possible to improve current standard and MDR treatments in order to diminish the likelihood of DR emergence while maintaining treatment effectiveness?

Response 3: Improving current regimens for the treatment of *M. tuberculosis* is the long-term goal of this research programme. To the best of our knowledge, our current manuscript is the first large-scale investigation into the collateral consequences of drug resistance in *M. tuberculosis*. We also provide evidence that combinations of drugs that exploit the collateral consequences of drug resistance have increased sterilizing properties and can reduce the emergence of drug resistance. In our original submission, we acknowledge various limitations of our current and provide avenues for future investigations to corroborate our findings and to expand this work into a preclinical space. We have expanded upon these concepts in the discussion of our revised manuscript.

See Discussion section. (Pages 22-23, lines 485-527):

*“Therapeutic targeting of collateral sensitivities has the potential to increase treatment efficacy and suppress the development of antibiotic resistance^{63,64}. Despite these promises, valid concerns remain regarding the conservation of collateral phenotypes across different genetic backgrounds and under different conditions⁶⁵. Amongst geographic lineages of *M. tuberculosis*, differences in genetic requirements can influence antibiotic susceptibility and evolvability of drug resistance⁶⁶⁻⁶⁸. Furthermore, compensatory mutations and combinations of resistance genotypes have been shown to alter the fitness costs of mono- and multi-drug resistant strains⁶⁹. Whilst collateral networks have been reported as being conserved amongst clinical isolates of *E. coli*⁸, determining whether these collateral pathways are conserved amongst diverse geographical lineages and multi-drug resistant backgrounds of *M. tuberculosis* requires further investigation. Importantly, this current study goes some way towards addressing these issues as we observed instances of collateral drug phenotypes being conserved amongst genetically diverse drug resistant strains. For example, a *katG* (i.e. INH), *atpE* (i.e. BDQ) and multiple *rpoB* (i.e. RIF and FIX) mutants all had increased sensitivity to growth inhibition by PA824, and numerous strains had increased sensitivity to the cytochrome *bc₁:aa₃* inhibitor Q203. Whilst the collateral drug sensitivity for the INH and BDQ resistant mutant against PA824 could be exploited within infected macrophages, the RIF resistant was more tolerant of killing by PA824 highlighting the need to determine how*

experimental conditions and metabolic state may alter the observed collateral sensitivities⁶⁵. Furthermore, the collateral sensitivity to PA824, did not alter the ability of *katG*, *atpE* and *rpoB* mutants to evolve secondary drug resistance. Whether this is the case for all collateral phenotypes, and whether the acquisition of multiple resistance mutations effects the stability of collateral phenotypes requires further investigation⁷⁰.

Drug cycling, in which drugs are switched throughout treatment to counter the evolution of resistance, is one strategy for exploiting collateral phenotypes and providing a personalised approach to infectious disease therapy. Mathematical models predicting the efficacy of drug cycling in a 'laboratory' and 'patient' model have highlighted the potential benefits of drug cycling⁶³. Comparable to prior drug-cycling experiments on other pathogens, our results demonstrate that collateral sensitivities can be targeted to select against drug-resistant strains of *M. tuberculosis* in both a homogenous and heterogenous population^{12,71}. Increasing advancements in *M. tuberculosis* diagnostics allows for resistance genotypes to be rapidly determined throughout the course of infection⁷². Combined with drug resistance in *M. tuberculosis* being restricted to spontaneous chromosomal mutations, rather than horizontal gene transfer, and *M. tuberculosis* having a relatively low mutation rate in vitro and in vivo, *M. tuberculosis* could be an interesting candidate for drug-cycling regimens^{73,74}. To achieve this there is a need to understand how drug cycling and prior drug resistance influences the evolution of secondary drug phenotypes in *M. tuberculosis*. Furthermore, whether drug cycling could be applied on a large scale and across a diverse array of health care settings as would be needed for the treatment of *M. tuberculosis* requires special consideration."

Comment 4: Regarding the Methods section, please detail a bit more how the variant calling step was performed after mapping, at the end of the "DNA isolation and WGS" section.

Response 4: We agree with the reviewer and have added more detail describing the variant calling component of our work in the revised manuscript. The below text is now in the DNA isolation and Whole Genome Sequencing section of the methods:

See Methods section: "DNA Isolation and Whole Genome Sequencing" (Pages 31-32, lines 690-728):

"Whole genome sequencing (Illumina 150 bp paired end reads) was performed at The Microbial Genome Sequencing Centre (Pittsburgh, USA). Resulting reads were paired and trimmed in Geneious Prime® 2020.2.4 using the BBDuk plugin. The *M. tuberculosis* mc²6206 parental strain was mapped to the H37Rv genome (Genbank accession number: NC_000962) using Geneious Bowtie-2 plugin to generate a consensus sequence. All subsequent *M. tuberculosis* mc²6206 resistant strains were mapped to the mc²6206 parental strain consensus sequence using Geneious Bowtie-2 plugin. SNPs and genomic variants were identified using the Geneious 'Find Variation/SNPs' built-in tool. Variants were called with a minimum coverage of 20, and a minimum variant frequency of 80%. All mutations identified in each drug-resistant strain and the variant frequency are described in Table S2. All raw Illumina reads are available at NCBI-SRA (project number: PRJNA914416)."

Reviewer #2 (Remarks to the Author):

This manuscript by Waller et al aims to study antibiotic cross-resistance (CR) and collateral sensitivity (CS) in *M. tuberculosis*. The phenomenon of CS, which has previously been examined in several studies going back 15 years or so, involves cases where the evolution of antibiotic resistance can result in increased sensitivity to unrelated drugs. A majority of the previous studies have examined this phenomenon in *E. coli* and *P. aeruginosa*. In this manuscript, the authors have performed a thorough study of CS (and CR) for 23 different drugs using an avirulent variant of *M. tb*. Using selection experiments combined with genetic and phenotypic tests they show in vitro and in macrophages that CS can eliminate/reduce the emergence of resistant mutants in the population. Overall, I think this is a well-designed and interesting study and the use of *M. tb* as the study organism, a major pathogen where resistance emerges exclusively by mutation, further increases its potential clinical value.

Comment 1: A key question regarding the potential clinical application of CS is how conserved it is between different isolates of *M. tb*? Even though the study was not designed to examine this it would be good if the authors would explicitly discuss this in the context of other species (e.g. *E. coli*) and the likelihood of this being a concern for *M. tb*. Personally, I think lack of conservation will be a major obstacle in utilizing CS clinically but perhaps *M. tb* is different because of its small pan-genome and limited genetic variability?

Response 1: We agree with the reviewer that determining if the identified collateral sensitivities are conserved across different resistance genotypes and geographical isolates is crucial for the ultimate clinical translation of this work. This is particularly important as prior work (see references PMID:31163081, PMID:34297925, PMID:23749189) has demonstrated that differences in genetic requirements between lineage 2 (i.e., HN878) and 4 (i.e., H37Rv) can influence antibiotic susceptibility and the evolution of drug resistance. Determining whether the collateral phenotypes identified in our current manuscript in an H37Rv derivative translate to other major geographical isolates, and whether the combination of resistance mutations in MDR/XDR backgrounds influences collateral phenotypes, are current lines of investigation.

Importantly, our study was able to demonstrate that collateral sensitization to PA824 was conserved amongst unique *katG*, *atpE* and *rpoB* mutants (RIF and FIX resistant mutants, FIX data below, 3 independent replicates), providing support for collateral phenotypes being conserved amongst unique drug-resistant genotypes. This data has been expanded upon in our revised manuscript and included as supplementary material (see Figure S3E).

Further to the reviewer's comment, prior work in *E. coli* has demonstrated that growth conditions and metabolic state can influence the observed collateral phenotypes (see PMID: 34846167). We have included additional results investigating if the PA824 collateral sensitivity of the INH, BDQ and RIF is translated to infected macrophages (see Figure 5D-F). Whilst both the INH and BDQ-resistant mutants are more readily cleared by PA824 compared to the parental strain, the RIF mutant had increased tolerance under these conditions (Updated figure 5 below).

Considering these results, the reviewer's comments, and comments from reviewer three below (see comment 2) regarding the possible influence of compensatory mutations, we have added the following to the discussion of our revised manuscript to highlight these potentially confounding factors. We have also updated the results of our manuscript to highlight this data:

See Discussion section. (Page 22, lines 485-509):

*“Therapeutic targeting of collateral sensitivities has the potential to increase treatment efficacy and suppress the development of antibiotic resistance^{63,64}. Despite these promises, valid concerns remain regarding the conservation of collateral phenotypes across different genetic backgrounds and under different conditions⁶⁵. Amongst geographic lineages of *M. tuberculosis* differences in genetic requirements can influence antibiotic susceptibility and evolvability of drug resistance⁶⁶⁻⁶⁸. Furthermore, compensatory mutations and combinations of resistance genotypes have been shown to alter the fitness costs of mono- and multi-drug resistant strains⁶⁹. Whilst collateral networks have been reported as being conserved amongst clinical isolates of *E. coli*⁸, determining whether these collateral pathways are conserved amongst diverse geographical lineages and multi-drug resistant backgrounds of *M. tuberculosis* requires further investigation. Importantly, this current study goes some way towards addressing these issues as we observed instances of collateral drug phenotypes being conserved amongst genetically diverse drug resistant strains. For example a *katG* (i.e. INH), *atpE* (i.e. BDQ) and multiple *rpoB* (i.e. RIF and FIX) mutants all had increased sensitivity to growth inhibition by PA824 and numerous strains had increased sensitivity to the cytochrome *bc₁:aa₃* inhibitor Q203. Whilst the collateral drug sensitivity for the INH and BDQ resistant mutant against PA824 could be exploited within infected macrophages, the RIF resistant was more tolerant of killing by PA824 highlighting the need to determine how experimental conditions and metabolic state may alter the observed collateral sensitivities⁶⁵. Furthermore, the collateral sensitivity to PA824, did not alter the ability of *katG*, *atpE* and *rpoB* mutants to evolve secondary drug resistance. Whether this is the case for all collateral phenotypes, and whether the acquisition of multiple resistance mutations effects the stability of collateral phenotypes requires further investigation⁷⁰.”*

Revised Figure 5: Collateral drug sensitivities are retained within infected THP-1 macrophages: (A-F) The survival of intracellular *M. tuberculosis* drug-susceptible or resistant strains within THP-1 macrophages was determined after 3 days of exposure to compounds at or above the MIC. MICs are as follows; Q203=0.05 μ M, BDQ=0.2 μ M, TAC=0.5 μ M, PA824=1 μ M. CFUs/mL were used to determine *M. tuberculosis* survival. Results are presented as percentage (%) survival relative to the no compound DMSO control. Line graphs represent the mean of three biological replicates, with individual replicate values plotted for each concentration. Results are from a representative experiment (n=2).

Comment 2: One weakness of all (?) studies of CS is that the conditions used to show that CS can reduce resistance evolution are often chosen to poise the system such that the expected effect can be seen. Thus, mutation supply rates (i.e. population size), drug level, time of exposure etc are typically not systematically varied to determine how these variables will affect the extent of CS. It would be useful if the authors discussed why the particular conditions were chosen and how altering them might change the ability of CS to reduce resistance evolution.

Response 2: Drug-resistant mutants can be isolated using a variety of approaches that expose bacteria to either progressively increasing or fixed concentrations of an antibiotic. Due to the long doubling of *M. tuberculosis* compared to other model bacteria such as *E. coli*, isolating resistant strains to increasing concentrations of antibiotic concentration are experimentally prohibitive. For this reason, we utilised an experimental approach in which populations of *M. tuberculosis* were exposed to fixed concentrations of an antibiotic. This method, which is analogous to a fluctuation assay, is consistent with the majority of the published literature investigating the in vitro drug resistance in *M. tuberculosis* (see references PMID: 23749189, PMID: 31721551, PMID: 32139546, PMID: 28760892, PMID: 24086479). These conditions also utilise large population sizes to isolate resistant mutants,

ensuring that there is sufficient diversity in the population to isolate rare mutations. Importantly, prior comparisons of adaptive evolution methods that vary population and exposure conditions when using *E. coli* as a model system have demonstrated that whilst there are subtle differences in the genetic and phenotypic outcomes, key adaptations are observed independent of selection protocol (PMID:28553265)

We agree with the reviewer that variations in population size and the duration and concentration of drug exposure can influence the collateral phenotypes that are observed. In the current study, we have investigated how variations in antibiotic concentration and exposure length influence the observed collateral phenotype (Figure 2 onwards). How variations in population size influence the collateral were not investigated but remain an ongoing line of investigation.

To address the comment of the reviewer, we have added the following statement to the methods section of our revised manuscript:

See Methods section: “Resistant mutant isolation” (Page 25, lines 553-559):

*“To isolate drug resistant strains of *M. tuberculosis*, we utilised an experimental approach consistent with prior studies investigating in vitro drug resistance in *M. tuberculosis*^{64,74,75}. Whilst alternative methods have been reported, the long doubling time of *M. tuberculosis* makes progressive exposure to increasing concentrations of antibiotics experimentally prohibitive when utilising multiple antibiotics. Importantly, comparative studies in *E. coli* have demonstrated that key adaptations are conserved across adaptive evolutionary protocols⁷⁶.”*

Comment 3. It would be useful if the authors state in the first section of the results (in text or supp table) for all the different clinically used drugs how representative the isolated in vitro mutants are (i.e. are they the mutations that are typically found in clinical isolates). Furthermore, state in text/table/fig (could for example be added in parenthesis next to the resistance mutations in fig 1) whether other mutations than the expected resistance mutations were present in the strains.

Response 3: We agree with the reviewer that this information would be useful. We added an additional information to table S2 (see response to reviewer 1 – comment 2), as well as adding asterisks to clinically relevant mutants in figure 1. We have also added the following to the results of our revised manuscript:

See Results section: “Collateral drug phenotypes of drug resistant *M. tuberculosis*” (Pages 6-7, lines 109-122).

*“For strains with resistance to clinically utilised anti-tubercular agents including rifampicin (RIF), fluoroquinolones (i.e. levofloxacin (LEV)), aminoglycosides (i.e. capreomycin (CAP), kanamycin (KAN) and streptomycin (STRP) and linezolid (LZD), our isolated mutations are consistent with those observed in clinical isolates of *M. tuberculosis*²². Strains with resistance to INH did not contain the dominant genotype (i.e. KatG^{S315T}) that is seen in the majority of clinical isolates²³. This is consistent with prior reports of in vitro isolated INH-resistant mutants²⁴. For some resistance loci that require inactivation to generate resistance (i.e. *ddn*, *fbjABC*, *rv0678*), available clinical data shows that there is a large spectrum of mutations²². Whilst most mutations identified in these genes in this study have not yet been observed clinically, the loss of function genotype does not preclude them from eventually being observed. Furthermore, some compounds have yet to be utilised in a clinical setting so clinically relevant resistance loci have yet to be defined. Secondary mutations identified in all strains are described in Table S2”.*

Comment 4. Fig. 1 is misleading with regard to the colours chosen to illustrate the extent of CR or CS. Since no cases of 9x or 27x MIC-fold changes were observed for CS I would remove them from the X-axis. This will also clearer emphasize that the big effects are very much skewed towards cross-resistance rather than collateral sensitivity.

Response 4: We agree with the reviewer's comments. We have removed the 27x and 9x collateral sensitization from figure 1.

Reviewer #3 (Remarks to the Author):

Waller et al. have investigated the development of collateral sensitivity effects in *M. tuberculosis*. They find multiple examples of collateral effects and show how these can be exploited to counter-select resistant isolates. This is an interesting and thorough piece of work (especially considering that it was performed in *M. tuberculosis*) that will be a worthwhile addition to the growing body of knowledge of drug resistance development in MTB. However, I do have some issues concerning the strong statements regarding the clinical potential of using collateral sensitive in a clinical setting. I think additional experiments are needed here and the conclusions should be toned down.

Major comments:

Comment 1: In the sections concerning treatment strategies (203-222 and 308-371), the authors choose sub-MIC antibiotic concentrations. In line 43-45 the authors write that ‘... patient non-compliance and suboptimal drug penetration frequently results in treatment failure, driving the evolution and spread of drug resistance...’. Keeping this in mind, is it (i) possible to achieve specific sub-MIC concentrations within the human body, and (ii) would this not result in even more resistance development (i.e. see <https://doi.org/10.1038/s41467-018-04059-1>)

Response 1: In our published study, we have used sub-inhibitory concentrations in experiments that involve drug combinations. This was done as part of proof-of-concept experiments to demonstrate the potential for drug combinations that target collateral drug phenotypes to select against the emergence of drug-resistant strains.

Specifically, experiments in figure 6 in our revised manuscript were designed to show that resistance to compounds at above inhibitory concentrations could be selected against by the presence of a partner compound. The partner compound was used at sub-inhibitory concentrations that would not affect the growth of the parental drug-susceptible strain, to demonstrate that the growth of the resistant mutant was selected against. If the partner compound was used at inhibitory concentrations, then there would have been no opportunity to observe a difference between the outgrowth of the WT and resistance mutant population. Furthermore, using antibiotics in combination at above inhibitory concentrations results in drug-drug interactions that alters their efficacy and would influence one’s ability to observe the impacts of targeting collateral drug phenotypes.

Our results in figure 4 E-L address the impacts of using above inhibitory concentrations against drug-resistant strains. In these experiments, collateral sensitivities could be targeted to select against drug-resistant strains in both homogenous and heterogeneous populations of cells. The results in Figure 4J-L are comparable to prior drug-cycling experiments that have demonstrated how switching antibiotic treatment can select against drug-resistant strains that are in a heterogenous population (see references PMID: 29307490, PMID: 24068739).

We agree with the reviewer and are not advocating for the clinical use of sub-therapeutic concentrations. However, others have suggested that targeting collateral sensitivities could be applied to drugs with narrow therapeutic windows. For example, in antibiotics that require a certain concentration to achieve bacterial killing, but higher doses are associated with increased toxicities, collateral sensitivity could be targeted to lower the dose required without decreasing efficacy (see reference PMID: 34584086).

To address the reviewers comments we have amended our revised manuscript as described below.

See Results section: “Collateral drug sensitivities can be exploited to impair the growth of drug resistant *M. tuberculosis*” (Page 11, lines 233-235):

“Strains were initially challenged with sub-inhibitory, rather than inhibitory concentrations, to highlight differences in drug sensitivity of resistant strains relative to the drug susceptible parent strain.”

See Discussion section (Page 21, lines 483-484):

We had altered the following sentence in the discussion as follows. “Our findings suggest that ~~when used at sub-inhibitory concentrations~~ TAC or potentially an alternative inhibitor of mycolic acid synthesis can actively select against INH-resistant mutants.”

Comment 2. Lines 188-202: The authors call these secondary mutations which might suggest that they appeared during the drug selection (next to the primary resistance mutations). From table S2 it is obvious that this is not the case. The identified mutations show that the strain stock has undergone an extensive multi-step adaptive evolution process (for example to adapt to the media or the auxotrophy mutations). I suggest that the authors (i) make a clear comment about this in the manuscript, and (ii) add a supplementary figure with the evolutionary trajectories of the clearly identifiable background mutations and mark out where in the tree the resistant mutations can be found. This will help to make the distinction where discrepancies in the collateral sensitivity profile are caused by the strain background and where due to the drug selection. Also, I suggest to isolate a new ‘wild-type’ strain that is adapted to media/auxotrophies for future work to reduce the diversity in the strain background.

Response 2: This was similar to the comment raised by reviewer 1 (comment 2). We have updated our supplementary information to include more information regarding secondary mutations (see revised Table S2).

Our resistant mutants were isolated using an experimental approach in which populations of *M. tuberculosis* were exposed to fixed concentrations of an antibiotic. This method, which is analogous to a fluctuation assay and is consistent with the majority of published literature investigating the *in vitro* drug resistance in *M. tuberculosis* (PMID: 23749189, PMID: 31721551, PMID: 32139546, PMID: 28760892, PMID: 24086479). We used the term “secondary mutations” in this manuscript to describe mutations that have occurred in the isolated strains but are not necessarily mutations associated with the observed phenotype. These secondary mutations would have appeared at various time points during the culturing of the parental strain prior to antibiotic exposure either before or after the acquisition of drug resistance genotypes. This explains why some drug-resistant strains with diverse resistant phenotypes have shared secondary mutations. The updated TableS2 includes additional information highlighting that secondary mutations were predominately conserved among resistant mutants when they were isolated from the same culture that was used in resistant mutant isolation studies (Table S2). In the majority of cases these resistant mutants that have shared secondary mutations each have distinct resistance/sensitivity profiles. See response to reviewer 1/comment 1 for a more detailed discussion of these secondary mutations.

The *M. tuberculosis* strain used in this study is a double auxotroph derived from H37Rv, a strain of *M. tuberculosis* that is routinely used in laboratories around the world. This strain, mc²6206 is avirulent even in immune-compromised SCID mice. This allows for mc²6206 to be used under BSL2/PC2 laboratory settings, rather than BSL3/PC3 containment. The auxotrophy of these strains is an essential requirement for them being used at BSL2/PC2 containment. Because of this we are unable to isolate a new “wild-type” strain that is adapted to the auxotrophy (See PMID: 29844114).

We have addressed these secondary mutations in our revised manuscript as described in our response to reviewer 1, comment 1 (see above).

Comment 3. Lines 270-273 and 362-365: If the collateral sensitivity phenotype causes no change in the development of new resistance, then where is the benefit of combining two drugs with collateral effects compared to two drugs without? In a clinical setting it is not feasible (or smart) to supply specific sub-Mic concentrations of antibiotics. The question that should be answered is if combining two drugs with a collateral sensitivity profile are better in preventing resistance emergence than two drugs without sensitivity. I don't think the current experiments actually show if this is the case.

Response 3: There are multiple benefits associated with the use of combination therapies, one of which is the ability of combination therapies to reduce the emergence of resistance (PMID: 36538813). There can also be benefits associated with drug synergy or as highlighted by our study collateral drug sensitivity that may increase the efficacy of combination therapies without influencing the evolution of drug resistance (PMID: 36538813). The two outcomes are mutually exclusive, and we don't believe that combination regimens with just one positive interaction preclude it from further consideration.

The experiments presented in our current study were designed to compare the potential benefits and differences between drug combinations that target collateral drug response. They were not designed to compare the outcomes of drug combinations with and without collateral drug responses. The reviewer is correct, that (i) the increased sensitivity of BDQ, RIF and INH mutants to pretomanid did not alter their ability to evolve secondary drug resistance and (ii) there was no change in the evolution of INH in the presence of 0.3x MIC of Q203 in frequency of resistance studies. However, it should be noted that 0.3x MIC of Q203 was able to delay the emergence of drug-resistant which may have unique benefits in a clinical setting.

Whilst we agree with the reviewer that sub-therapeutic concentrations should not be used unless clinically justified, others have suggested that collateral sensitivities could be applied to drugs with narrow therapeutic windows. For example, in antibiotics that require a certain concentration to achieve bacterial killing, but higher doses are associated with increased toxicities, collateral sensitivity could be targeted to lower the dose required without decreasing efficacy (PMID: 34584086).

Minor comments:

Comment 4. According to reference 1, the first-line treatment for MTB includes pyrazinamide and ethambutol. It seems like pyrazinamide was not included in the study and resistance mutants for ethambutol were removed due to slow growth. This is unfortunate since it eliminates half of the first-line drugs out of the study. It would be important to know if the current treatment regimen already uses collateral sensitivity effects or not.

Response 4: We agree that it is disappointing we were unable to include PZA or EMB-resistant mutants in this study. As noted by the reviewer, the EMB-resistant mutant grew poorly in 96 well plates, which complicated their use in our assays. PZA is complicated to use in laboratory settings because it is generally inactive at neutral pH and requires an acidic or basic environment for activity. There is evidence to suggest PZA can be active at neutral pH, but this activation is temperature dependant (PMID: 27270287). This need for unique culturing conditions would have complicated our experimental setup which involved the screening of 48 strains against 23 antibiotics.

Furthermore, the majority of multi and extensively drug-resistant strains contain mutations that provide resistance to RIF, INH, fluoroquinolones, and aminoglycosides all of which are included in our current study. Because of this, we don't believe that the lack of PZA and EMB-resistant mutants weakens the findings of this study.

Comment 5. Lines 106-110: I don't see an example of more than three mutants for a single drug in figure 1 so I would drop the 'at least' from the sentence.

Response 5: This has been removed from our revised manuscript.

Comment 6. Lines 110-119: If a 3-fold dilution series was prepared, then how do you measure a 2-fold change? I suggest to remove the 2-fold changes from the results since they should not be considered significant. Having the additional shade of blue in figure 1 makes it harder to see what is actually significant.

Response 6: Our initial experiment was designed to include a 3-fold dilution curve to cover a wide range of drug concentrations. This was done as we were unsure what magnitude of changes in drug susceptibility we would observe. MICs were determined from the slope and inflection point in the Gompertz equation, with the resulting MICs for drug-resistant strains compared to the drug-susceptible parent that was included in each experiment.

We agree with the reviewer that the experimental setup was not designed to detect 2-fold MIC changes. This limitation was reported in our initial manuscript. However, because there were a large number of strains that had a reproducible 2-3-fold shift in MIC across >4 replicates we included this information and noted these strains as having low-level changes in susceptibility. Importantly, several of these 2-3-fold shifts in MIC are consistent with the prior results, e.g. low-level resistance of *mshA* and *mshC* mutants against INH. Furthermore, we were also able to validate the novel low-level cross-resistance between PA824 resistance loci and clofazimine. We believe that these points warrant the inclusion of data reporting 2-3-fold MIC shifts for validation in future studies. We agree with the reviewer and have amended figure 1 to show a greater level of shading between the 2-3 and >3-fold shifts in MIC phenotypes (see revised Figure 1).

Comment 7. Figure 1: The shades are too similar, especially if the non-significant 2-fold changes are included. I can honestly not tell if any of the drugs have 9-fold reduction in MIC or not. In the same direction, a 27-fold change is given a colour but it doesn't seem to exist in the data set.

Response 7: This comment was raised by reviewer 2, comment 4 and we have addressed as discussed above.

Comment 8. How do the isolated mutations compare to the once found in clinical MTB isolates?

Response 8: This comment was raised by reviewer 2 (see comments 1 and 3). Please see responses above.

Comment 9. For rifampicin only a single *rpoB* mutation was isolated (*rpoB* S450L). This is the most common clinical mutation in MTB but there are many others that are clinically relevant. Do these also show collateral sensitivity effects? Maybe it would be possible to isolate at least a couple of the other relevant mutations and test for sensitivity to PA824. Also *rpoB* S450L often with secondary compensatory mutations in *rpoA*, *rpoB*, or *rpoC*. Could that effect the collateral sensitivity profile?

Response 9: S450L is the dominant mutation for RIF resistance in clinical settings, with evidence from a high-burden country suggesting around 56% of RIF-resistant isolates harbour the SNP (PMID: 34641802).

In our original submission, a strain with resistance to fidaxomicin (RpoB^{Q1080R}) had a low-level increase in sensitivity to PA824 (figure 1). Data from 3 independent replicates are included below. Whilst the mutation is not associated with resistance to rifampicin and is not seen in clinical populations of *M. tuberculosis*, it supports the observation that diverse mutations in RpoB may be associated with altered sensitivity to PA824. This data has been expanded upon in our revised manuscript and included as supplementary material (see Figure S3E).

We agree with the reviewer that determining whether collateral phenotypes are conserved across different mutations in the same gene and in the context of different genetic backgrounds (i.e., compensatory mutations and MDR strains) is essential. We have expanded upon this in the revised manuscript. See response to reviewer 2 (comment 1) above.

REVIEWERS' COMMENTS

Reviewer #3 (Remarks to the Author):

I am satisfied with the authors changes and replies.